# Accurate Neural Network Pruning Requires Rethinking Sparse Optimization

**Denis Kuznedelev** [*]                                   *denis.kuznedelev@skoltech.ru*
*Skoltech & Yandex*

**Eldar Kurtic**[*]                                            *eldar.kurtic@ista.ac.at*
*IST Austria*

**Eugenia Iofinova**[*]                                      *eugenia.iofinova@ista.ac.at*
*IST Austria*

**Elias Frantar**                                              *elias.frantar@ista.ac.at*
*IST Austria*

**Alexandra Peste**                                        *alexandra.peste@ista.ac.at*
*IST Austria*

**Dan Alistarh**                                              *dan.alistarh@ista.ac.at*
*IST Austria & Neural Magic*

**Reviewed on OpenReview:** *https://openreview.net/forum?id=vgthYeRBAF*

## Abstract

Obtaining versions of deep neural networks that are both highly-accurate and highly-sparse is one of the main challenges in the area of model compression, and several high-performance pruning techniques have been investigated by the community. Yet, much less is known about the interaction between sparsity and the standard stochastic optimization techniques used for training sparse networks, and most existing work uses standard dense schedules and hyperparameters for training sparse networks. In this work, we examine the impact of high sparsity on model training using the standard computer vision and natural language processing sparsity benchmarks. We begin by showing that using standard dense training recipes for sparse training is suboptimal, and provide evidence that this results in *under-training*, loosely defined as using a suboptimal number of passes over the training data. We present training recipes for mitigating this issue for both sparse pre-training of vision models (e.g. ResNet50/ImageNet) and sparse fine-tuning of language models (e.g. BERT/GLUE), achieving state-of-the-art results in both settings in the high-sparsity regime, and providing detailed analyses for the difficulty of sparse training in both scenarios. Our work sets a new benchmark in terms of the accuracies that can be achieved under high sparsity, and should inspire further research into improving sparse model training, to reach higher accuracies under high sparsity, but also to do so efficiently.

## 1 Introduction

The difficulty of finding deep neural networks (DNNs) that are both *accurate and sparse*, i.e., closely match the accuracy of dense models while having a large majority of their weights set to zero, is one of the main

---

[*]These authors contributed equally. Author order was determined by experimental load (highest first).

challenges in the area of model compression. On the conceptual side, this challenge connects to fundamental questions related to the *Lottery Ticket Hypothesis (LTH)* (Frankle & Carbin, 2019; Frankle et al., 2019), which posited that such sparse masks exist, and that, in some cases, they can even allow accurate training of sparse models *from scratch*, that is, applying the sparsity mask at initialization. On the practical side, obtaining highly-sparse and accurate networks can lead to significant practical speedups, both for inference (NeuralMagic, 2022) and training (Nikdan et al., 2023).

In this work, we focus on the challenge of obtaining accurate DNNs in the high-sparsity regime, and investigate the barriers to obtaining **highly-sparse** and **highly-accurate** variants of DNNs for standard vision and language tasks. We mainly focus on two tasks that are, arguably, the standard benchmarks for sparsity in vision and language, respectively: image classification using the ResNet50 model (He et al., 2016) on the ImageNet-1K dataset (Russakovsky et al., 2015), e.g. Hoefler et al. (2021); Dong et al. (2017); Gale et al. (2019); Evci et al. (2020); Singh & Alistarh (2020); Savarese et al. (2021); Peste et al. (2021), and language modelling using the BERT-base model (Devlin et al., 2019) on the GLUE benchmark datasets (Wang et al., 2018), e.g. Sanh et al. (2020); Hoefler et al. (2021); Kurtic & Alistarh (2022); Kurtic et al. (2022). Roughly, for both benchmarks, it is known that sparsities lower than 90% can be achieved with approximately 1% accuracy loss relative to the original dense model, but accuracy rapidly decreases in the 90-95% range (Hoefler et al., 2021; Evci et al., 2020), and that decreases are drastic at higher ($\geq 95\%$) sparsities (Singh & Alistarh, 2020; Kurtic et al., 2022). In this paper, we investigate the reasons behind this accuracy loss due to sparsity, mainly targeting *high sparsity*, i.e. sparsities between 90% and 99%, studying the difficulty of obtaining accurate models in this range, and providing ways to circumvent it.

**Contribution.** We begin from the observation that, when training sparse models from scratch, following standard *dense training* schedules, *sparse models show clear evidence of undertraining*: both their accuracy and loss fail to saturate under standard number of training epochs, and their output continues to have high entropy. This finding suggests that maximization of the accuracy of sparse models requires longer training than the dense optimization recipes adopted in most of the work on model sparsification.

Motivated by this observation, we propose a combination of techniques which can mitigate the inherent difficulty of sparse training. As a consequence, we significantly improve on the best currently-known sparsity-accuracy trade-offs on standard sparsity benchmarks for both image classification and language modelling. Specifically, we consider the two classic sparsification benchmarks in this setting: image classification (ResNet50 on ImageNet) and language modelling (BERT or SQuAD and GLUE tasks), and set new state-of-the-art results in both settings.

For image classification, we obtain, for the first time, highly-accurate sparse versions of ResNet50, such as a 90%-sparse model with 78.5% Top-1 accuracy, a 95%-sparse model with 77.7% Top-1 accuracy, and a 98%-sparse model with 75.2% Top-1 accuracy. In the same context, the highest accuracy for a dense model we could obtain is 78.78% Top-1. In addition, we show that stable results can be obtained even for extreme sparsities (e.g., 99%).

We also extend our results to language models from the BERT family Devlin et al. (2019), where we show that on challenging modeling tasks, as measured by the drop in accuracy relative to the dense model, similar techniques can improve results by 3 points in accuracy relative to the current state-of-the-art results at 90% sparsity. We arrive at these results as follows:

- We perform an analysis of the output and training characteristics of models trained using current state-of-the-art techniques, relative to their dense counterparts. First, we show that sparse DNNs obtained via many current techniques behave similarly to dense models that have been *undertrained*, i.e. executed for a sub-optimal number of epochs: specifically, they tend to have high output entropy (alternatively, low "output confidence"), which correlates with their reduced accuracy.

- This analysis provides clear evidence that optimizing *sparse models* is more difficult than standard *dense* optimization (Evci et al., 2019). This observation stands in contrast to the fact that most current sparsification techniques use standard *dense* training recipes for fine-tuning and recovery. We exploit this insight to obtain state-of-the-art accuracy for sparse models in two popular scenarios:

*sparse pretraining*, i.e. training sparse models from scratch, and *sparse transfer*, i.e. optimizing a sparse pretrained model onto a target transfer task.

- In the *sparse pretraining* scenario, illustrated by the standard task of obtaining a highly-sparse ResNet50 model on the ImageNet dataset, we show that we can circumvent the difficulty of sparse training by adopting a variant of the Alternating Compressed/Decompressed (AC/DC) algorithm (Peste et al., 2021) for training sparse DNNs, which has convergence guarantees for sparse recovery. Specifically, we show that, by scaling the algorithm's runtime, we can obtain state-of-the-art results for sparse pretraining on ImageNet for ResNet50 and MobileNet models, and reach extremely high sparsities (e.g. 98% and 99%) while still obtaining stable convergence. Moreover, only sparse models benefit from extended training, whereas dense models start to overfit with longer training.

- We complement our analysis with a study of the *sparse transfer* scenario, popular in language modeling. Here, the difficulty of sparse training can manifest itself through both *undertraining* and *overfitting*, depending on the parametrization of the chosen transfer learning recipe, specifically on the training length. We address this via a modified version of the *gradual layer unfreezing* approach (Howard & Ruder, 2018), tailored towards a *sparse* transfer learning scenario, which allows us to obtain state-of-the-art results in the case of BERT-base transfer on downstream datasets.

**Discussion.** Overall, our results suggest that the difficulty of obtaining highly-accurate sparse models is closely linked to the difficulty of accurate sparse optimization using current state-of-the-art techniques. Specifically, our work improves the best known results on standard sparsity benchmarks, for both sparse pretraining and sparse finetuning, both in terms of absolute accuracy, and accuracy loss relative to the dense baseline. Moreover, we observe the following:

- Achieving state-of-the-art sparsity-vs-accuracy trade-offs currently requires using significant additional computational complexity and more epochs for training the sparse models, relative to the best known dense training methods. In turn, this suggests that sparse optimization may be inherently harder than its dense counterpart.

- Reaching high validation accuracy for sparse models is strongly linked to reaching low training loss, which occurs at a slower rate for sparse models in the case of SGD-based optimization. At the same time, we do observe overfitting behavior (decrease of validation accuracy w.r.t. increased training time), especially at lower sparsities.

- To further investigate the hardness of sparse optimization, we perform an analysis of the loss landscape of accurate sparse networks both in terms of sharpness and loss interpolation / mode connectivity. We observe that achieving highly-accurate sparse networks from initialization requires overcoming multiple loss barriers, and that sparsity mask exploration may be a key ingredient for overcoming these barriers.

- In addition, we investigate the relationship between standard hyperparameters such as weight decay, on the one hand, and sparsity structure, on the other. We find that careful setting of weight decay is critical for accurate sparsity, and that weight decay additionally induces (partial) structured sparsity in highly-sparse models. This provides a first explanation to the emergence of structured sparsity in unstructured sparse networks, which has been observed previously (Peste et al., 2021; Iofinova et al., 2022; Yin et al., 2023).

In summary, our results set new accuracy thresholds for sparse models using relatively simple techniques, and can be reproduced in reasonable time on commodity hardware. As such, they should serve as motivation for the community to investigate improved *sparsity-aware* optimization techniques, specifically allowing for faster, more efficient accuracy recovery.

## 2 Background and Motivation

Formally, accurate pruning is a constrained optimization problem which, given the objective of minimizing a loss function $\mathcal{L}$, aims to find an "optimal" sparsity mask $\mathbf{M}^\star$ with a given target sparsity $s$, fraction of zero parameters,[1] and weights $\mathbf{W}^\star$ such that

$$\mathbf{M}^\star, \mathbf{W}^\star = \mathrm{argmin}_{\text{mask } \mathbf{M}, \text{ weights } \mathbf{W}} \left[ \mathcal{L}(\mathbf{M} \odot \mathbf{W}) \right] \quad \text{with } \mathrm{nnz}(\mathbf{M}) \leq (1-s)\mathrm{numel}(\mathbf{M}). \tag{1}$$

In its general form, where both the optimal mask and the optimal weights must be determined, this question is NP-hard (Blumensath & Davies, 2008), even for simple least-squares loss. However, this problem can be made tractable if we assume a fixed mask, or we wish to approximate the sparsity of the mask, e.g. Axiotis & Sviridenko (2020).

In the context of pruning, this procedure can be logically split into 1) determining the sparsity mask $\mathbf{M}$, which is often separated from 2) the optimization procedure over the non-zero weights. For instance, the standard Lottery Ticket Hypothesis (LTH) approach (Frankle & Carbin, 2019; Chen et al., 2021b) is to first identify a "ticket" mask by performing weight selection by magnitude over an already-trained model, followed by SGD-based finetuning, using the initialization and the same set of hyperparameters as for dense training.

While several novel ways of choosing or updating the sparsity mask choice (step 1), have been investigated, by and large, for the second step, that of optimizing the remaining weights, sparse training methods largely emulate the hyperparameters of the baseline dense model, including the total number of training epochs (Gale et al., 2019; Jayakumar et al., 2020; Evci et al., 2020; Peste et al., 2021). However, it is intuitive that the problem of simultaneously finding near-optimal weights and a near-optimal mask may be harder to solve than a standard dense loss minimization problem.

This naturally motivates an in-depth investigation into the following questions: *can optimization over sparse networks converge with the same rate as over dense ones?*, and *are dense training recipes well-suited for sparse training?* In this paper, we provide evidence that the answer to both questions is *negative*, suggesting that improved optimizers may be required for obtaining accurate sparse models under reduced training budgets.

## 3 Related Work

The goal of most sparsification methods (Hoefler et al., 2021) is to create a DNN that is as accurate as possible, while maximizing sparsity. This goal can be achieved via different strategies: for instance, *post-training sparsification methods* assume a *pretrained dense model*, from which weights are removed either in a single step (one-shot) or progressively (gradual pruning). By contrast, in *sparse training methods*, parameters are pruned from the model during training from scratch, either close to initialization (Evci et al., 2020; Jayakumar et al., 2021; Lee et al., 2019; Vanholder, 2017; Schwarz et al., 2021), or progressively as the model is trained (Han et al., 2015; Gale et al., 2019; Savarese et al., 2021). A subset of sparse training methods are *dynamic*, in the sense that weights may be reintroduced during training (Evci et al., 2020; Peste et al., 2021).

In this work, we mainly focus on the *high-sparsity regime*, in which *sparse training* methods provide the best known accuracy-vs-sparsity trade-offs. We begin by discussing methods for computer vision. Here, Gradual Magnitude Pruning (GMP), in which the lowest-magnitude weights are progressively removed throughout training, is a common baseline. In Gale et al. (2019), GMP was shown to be competitive with more sophisticated pruning methods on image classification models when properly tuned; similar results were later shown for language models (Kurtic & Alistarh, 2022).

The RigL pruning method (Evci et al., 2020) is a common, high-performing benchmark for dynamic sparse training. In this method, the weights are initially pruned to the target sparsity and trained through (sparse) stochastic gradient descent. Periodically, however, the mask is updated by selecting weights with the highest magnitude gradient, subject to a limit on the total mask change. The authors run this method using two sparsity targets - Uniform sparsity, where all layers (except the first and last) are pruned to the same

---

[1]A *sparsity mask* is simply a binary tensor of the same dimensions as the model, with 0 at the indices of the sparsified entries, and 1 at the other indices.

proportion, and Erdős–Rényi Kernel (ERK), where layer sparsity targets are set to optimize performance. The authors test their method in the normal-schedule (100 epochs on Imagenet) and 5x training regime, getting results of 73.22% validation accuracy and 74.63% validation accuracy at 95% global (ERK) and uniform sparsity, respectively when training for 500 epochs. Extending training to 10 000 epochs (100x) further allowed the authors to produce 99% sparse (ERK) ResNet50 models with 68.5% accuracy on ImageNet. RigL was improved by combining it with ITOP (Liu et al., 2021), by altering training hyperparameters to encourage mask exploration, which was shown to improve RigL results at medium (80-90%) sparsity (see Table 1).

The GraNet(Liu et al.) method extends this approach by making it gradual - either starting from a dense network and performing RigL-like updates while simultaneously increasing sparsity until the target sparsity is achieved, or by starting by a partially sparse (50%) network and doing the same. Models trained with the sparse-init version of GraNet achieved 72.3% validation accuracy at 95% global sparsity when training for 100 epochs.

The AC/DC pruning method (Peste et al., 2021) alternates dense and sparse pruning phases of several epochs each, effectively co-training dense and sparse models. Similar to RigL, AC/DC was tested in the normal and extended training regime, creating 95% globally sparse ImageNet-1K ResNet50 models with 73.14% top-1 accuracy, and 68.44% top-1 accuracy 98% sparse models after 100 epochs of training. The authors also experiment with extended training times, producing 95% uniform sparsity ResNet50 models with 74.3% validation accuracy.

Another successful pruning approach is the combination of Powerpropagation (Schwarz et al., 2021) with Top-KAST (Jayakumar et al., 2021). In Powerpropagation, the weights are reparametrized using $f(w) = w|w|^{\alpha-1}$ for $\alpha > 1$, effectively encouraging high-magnitude weights to continue increasing while lower-magnitude weights are driven toward 0. Top-KAST is a dynamic sparse training scheme that is largely similar to RigL: in Top-KAST, for a target density $D$, the gradients of the top $D' < D$ weights are computed in each backpropagation round and allowed to accumulate, and the masks at these respective sparsities are periodically recomputed. The combination of these two methods results in 77.16% accuracy at 90% sparsity when trained for 3x their baseline of 32K steps.

The recently-proposed ST-3 method (Vanderschueren & Vleeschouwer, 2023) uses the technique of soft thresholding with straight-through gradient estimation to progressively prune neural networks while allowing weights to move more smoothly between the dense and sparse states. Using this method, the authors were able to achieve ImageNet accuracies of between 74% and 75% at 96% sparsity on ResNet-50, depending on the method variant used.

Additionally, some works have explored the difficulty of sparse optimization (Evci et al., 2019), explored changes to dense training pipelines to improve sparse training (ab Tessera et al., 2021; Jaiswal et al., 2022), or focused on the creation of sparse accurate neural networks outside of the standard paradigm of simultaneously searching for the optimal mask and weights. Notably, (Liu et al., 2021) explored the impact of mask exploration (that is, the total number of explored parameters at any point in sparse training), demonstrating the positive effect of extended training on both sparse network performance and total number of explored parameters. The STEP (Lu et al., 2023) learning method explored the interaction of sparsity with the Adam optimizer (Kingma & Ba, 2015), finding that the masked weights lead to an incorrect estimate of the second moment during optimization; these observations led to their proposal of a new method for N:M sparsity that alleviates these effects. The GradMax method (Evci et al., 2022) initializes a small neural network, then uses predicted gradients to grow a larger (while still small) neural network by adding additional neurons.

The problem of the sparse optimization also emerges in the context of the optimal transport (Peyré & Cuturi, 2020; Cuturi, 2013). It is often desirable to have a sparse assignment between the source and target domain. Several works have studied this question with applications to color transfer (Blondel et al., 2018) and sparse mixture of experts (Liu et al., 2022).

**Language models** For language models, the standard compression pipeline consists of two stages: pre-training on a large unlabeled text corpus followed by fine-tuning on a small and labeled task-specific dataset. The former is used to capture the statistical patterns and relationships that exist in the natural language,

allowing the model to recognize and even generate various linguistic patterns. The latter stage, fine-tuning on a downstream task, builds on top of the learned representations and adapts them to solve specific tasks such as text classification, sentiment analysis, duplicate detection, etc. Sparsity has been explored in both stages: pruning during pre-training and pruning during fine-tuning.

Methods such as Movement Pruning (Sanh et al., 2020) and The Optimal BERT Surgeon (oBERT) (Kurtic et al., 2022) make use of first-order (gradient) and second-order (curvature) information, respectively, to guide pruning decisions during the fine-tuning stage. However, recent work observed two problems with this approach when applied on small datasets: (Zhang et al., 2022) demonstrated instability due to large variability of estimated importance scores, while (Huang et al., 2021) observed overfitting despite reduced expressive power due to pruning. From the practical side, this approach is less favorable for practitioners as it requires extensive pruning-domain knowledge to properly configure pruners for each model and dataset combination. Therefore, the main focus of our work is on the other stage, leveraging already sparse pre-trained models with transfer learning to obtain highly accurate task-specific fine-tuned models. Prune Once for All (Prune OFA) (Zafrir et al., 2021) and oBERT (Kurtic et al., 2022) represent the most recent state-of-the-art techniques addressing this problem. Both methods first prune the model during the pre-training stage, and then apply transfer learning with a fixed sparsity mask to obtain fine-tuned and sparse models on various downstream datasets.

**Impact of sparsification beyond top-1 accuracy**   An open area of research is the impact that pruning in general, and the choice of pruning method in particular, have on the resulting model. In particular, pruned models have been shown to be more vulnerable to bias (Hooker et al., 2019; 2020; Iofinova et al., 2023), and worse at prediction accuracy under distribution shift (Liebenwein et al., 2021). Recent works by (Chen et al., 2021a) and (Iofinova et al., 2023) investigate the effects of pruning on a range of model trustworthiness metrics and find mixed results, with sparse neural networks having better calibration, but exaggerating spurious patterns in the existing data. Finally, works such as (Iofinova et al., 2022) and (Chen et al., 2021b) investigated the capacity of sparse CNNs for domain adaptation via transfer learning, finding that sparsely trained networks can have more generalizable features than dense ones.

## 4   The Difficulty of Sparse Pretraining of Vision Models

### 4.1   Sparse Vision Models Show Evidence of "Undertraining"

We begin by investigating correlations between the performance and output characteristics of dense and sparse models trained for increasing number of epochs. Specifically, we examine two key metrics: *Top-1 accuracy* on the validation/test set, and the *loss on the train set* for the trained models, while scaling the number of training epochs and the associated hyperparameters correspondingly.

We will examine the evolution of these metrics as we increase the number of epochs, in parallel for sparse and dense models. We specifically look out for instances where sparse models behave similar to dense ones that have been trained for a sub-optimal (too low) number of epochs, a phenomenon we simply call *undertraining*.

**Metrics: Output Loss and Entropy.**   We examine model fit to the training data via the training loss at the last epoch of training. For multiclass classification, traditionally cross-entropy loss is used. We compute the cross-entropy loss by taking the softmax over the vector of output values of the network and then applying the standard cross-entropy formula, where the cross-entropy is taken with respect to the correct label distribution for the model (1 for the correct class and 0 otherwise). For an output of a network outputting a vector $Z = (z_1, z_2, ..., z_C)$ of size $C$ with correct label $L$, cross-entropy $CE$ is given by the following formula:

$$CE(Z) = -\log\left(\frac{e^{z_L}}{\sum\limits_{j=1}^{C} e^{z_j}}\right). \tag{2}$$

Intuitively, the loss of the model is related to its "confidence" in the correct predictions, or equivalently could be said to measure the model's fit to the training data.

We use this quantity as it is conventional with respect to measuring model convergence; however, we consider the *entropy* computed over *test* data to be an equally good choice, as it captures the model's confidence in its predictions (whether they be correct or incorrect) and can be computed on a *test* set, without access to the correct labels. We show in Appendix C that the two metrics give nearly identical results in our experiments.

We expect a sufficiently large and well-trained model to have low loss on the training data. However, as is conventionally known, continued training on dense and low-sparsity models can result in in overfitting will lower these metrics further. Here we investigate whether the same rule applies to models with higher sparsity.

**Experimental setup.** We examine validation accuracy on trained sparse and dense ResNet50 models on the ImageNet-1K dataset and compare it to the train loss on the last epoch of training. All models were trained using standard hyperparameters (see Appendix A) except for the difference in number of training of epochs in different experiments. Measurements represent the final accuracy and training loss after the last training epoch, so each marker on the plots represents a full experiment, rather than an intermediate checkpoint. Sparse models were pruned with Alternating Compression/Decompression (AC/DC) (Peste et al., 2021), likewise adjusting the total number of compressed and decompressed phases to the total run length. AC/DC was chosen as it was among the best-performing methods across all sparsities and training lengths (see Section 4.2.1). We use the FFCV library (Leclerc et al., 2022) for fast loading of the data. In contrast with other runs presented in this paper, we do not use progressive resizing or label smoothing, as the latter explicitly encourages high prediction entropy and cross-entropy. In these experiments, we keep the first and last layer dense.

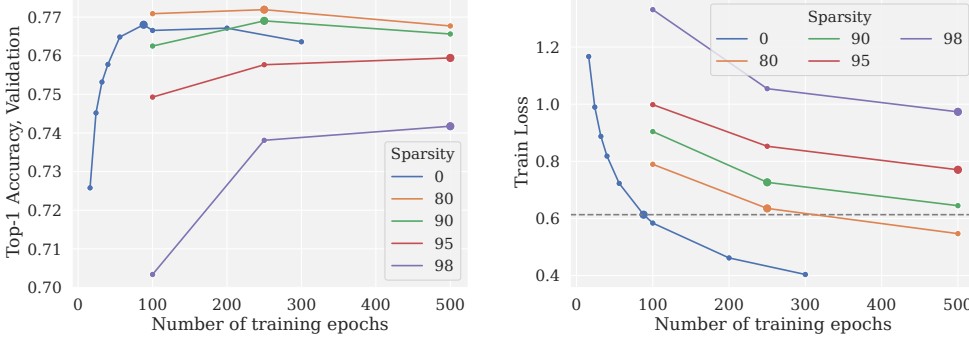

Figure 1: Average validation accuracy (left), and Train loss at final epoch (right) for sparse and dense ImageNet models trained for different numbers of epochs. The highest-accuracy model for each sparsity level is highlighted with a larger marker. The cross-entropy loss and entropy level of the dense model is also shown with a dashed line, to simplify comparison.

**Results.** Our results are presented in Figure 1. On the left panel, we show the top-1 accuracy of the final models. We observe that 80% and 90% sparse models reach an accuracy that is similar to dense models, even slightly exceeding dense accuracy at 80% sparsity. Accuracy drops at higher sparsity (95% and 98%); this is consistent with the original AC/DC paper and results from other pruning methods. Examining accuracy across epoch budgets, and focusing on the best-performing model for each sparsity level, we observe the following:

- *The dense model requires the fewest epochs* (88) to reach its best validation accuracy, and extending the training recipe results in *worse performance for the dense model*, commonly known as "overfitting."

- *The outcome changes if we examine sparse models*, for which the ideal training length increases with sparsity: 250 epochs for 80% and 90% sparse models, and at least 500 epochs—the longest schedule

we tried in this experiment—for 95% and 98% sparse models. Even at 500 epochs, the accuracy increase/loss decrease for these models does not appear to be saturated.

We now examine loss on the training dataset in more detail. We observe that the training loss always decreases when the number of training epochs is increased. However, sparse models trained for the standard 100 epochs show similar training loss to dense models trained for far fewer epochs. For example, dense models trained for 24 epochs have a similar training loss to 95% sparse models trained for 100 epochs, while dense models trained for 100 epochs have a slightly lower training loss than 80%-sparse models trained for 250 epochs. When we consider the best-performing models at their respective sparsity levels, we find that they have similar training loss to the top-performing dense model, in cases where such low loss/entropy can be achieved in a reasonable number of epochs (at 80% and 90% sparsity); at all sparsities. Further, continuing to train sparse models to until training loss drops below the training loss of the optimal dense model results in worse validation accuracy (overfitting).

**Discussion.**  These findings further support our hypothesis that, due to the inherent difficulty of sparse optimization, using standard training recipes is not sufficient for sparse training, and suggests that longer training may mitigate this effect. Further, results suggest that training loss can act as a useful criterion to validate that the sparse models are properly trained[2], with the latter criterion being also useful in cases where access to train data, or to any labeled data, is not possible.

In Appendix Section C, we consider the alternative Validation entropy metric, and present a similar validation on the Celeb-A dataset.

## 4.2   State-of-the-Art Accurate Sparse Pre-Training on ImageNet

The above observations for vision models suggest that successful sparse training may benefit from an extended training schedule. We now build on this idea to achieve state-of-the-art results for the classic ResNet50/ImageNet benchmark by using an extended-training version of AC/DC, which we call AC/DC++.

### 4.2.1   Comparing Sparse Training Methods

For the following experiments, we start from the current state-of-the-art training approach for ResNet50/ImageNet training, using the Pytorch FFCV package (Leclerc et al., 2022). In addition to an extended training schedule, we use label smoothing and a linear learning rate decay with warm-up, as well as progressive resizing of input samples [3]. In this context, we implemented three leading sparse training methods: Gradual Magnitude Pruning (GMP) (Zhu & Gupta, 2017), RigL (Evci et al., 2020) and AC/DC (Peste et al., 2021), which we execute for an increasing number of epochs between 100 (standard) and 1000 (10x). For this, we scale the original training schedule proportionally, following the proportions employed by the original methods. For this experiment, models are compressed to 80%, 90%, and 95% sparsity. Following the most common experimental setup, we prune all weights in convolutional and linear layers (including input convolution and classification head). The exact training recipe is presented in detail in Appendix A. We note that each of the experiments presented in the paper takes less than a day on a standard 8-GPU NVIDIA RTX 3090 server. The results, in terms of accuracy and loss vs number of training epochs are presented in Figure 2 for 95% sparsity and in Figure P.9.

**Results.**  The results show a strong correlation between how well the methods achieve reduction in loss and their validation accuracy. This reinforces the point that sparse training methods saturate slower, both in terms of training loss and validation accuracy. This has also been investigated by prior work: Gale et al. (Gale et al., 2019) found that extended training did improved results for GMP in some cases, while RigL (Evci et al., 2020) and Powerpropagation (Schwarz et al., 2021) found diminishing improvements. At the same time,

---

[2]The 98% sparse model will likely never reach the entropy of the optimal dense model, suggesting that the accuracy may continue to improve with very long training schedules. In fact, the authors of RigL trained a 99% sparse model for 100 times the dense training time and were not able to saturate its accuracy. See `www.github.com/google-research/rigl#extended-training-results`.

[3]We follow the setup from the FFCV ImageNet example repository for ResNet50.

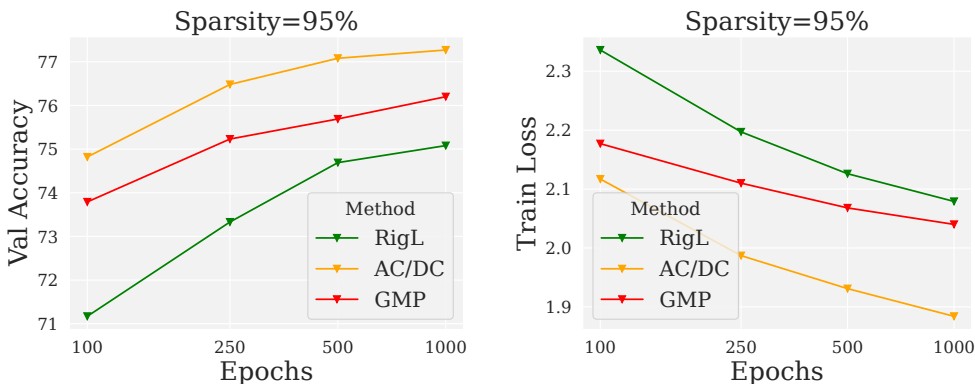

Figure 2: (**left**) Validation accuracy on ImageNet-1k vs number of epochs for different sparse training methods. (**right**) Training loss on ImageNet-1k vs number of epochs for different sparse training methods.

we notice a significant difference between methods: specifically, AC/DC starts at a slightly better accuracy point, and consistently outperforms other methods both in terms of loss achieved, and in terms of validation accuracy, as we increase training time. (This is consistent with the AC/DC original results, executed at 100 epochs (Peste et al., 2021).) We observe that this correlates with the theoretical computational cost (FLOPs) of the methods: AC/DC will use more FLOPs than other methods due to the dense training phases, while GMP uses more FLOPs than RigL due to gradually increasing sparsity. In turn, this could also be correlated with the amount of mask exploration performed by the algorithm during training. At low sparsity RigL performs slightly better than GMP, but for higher sparsity GMP appears to perform better. For the smallest 80%, 90% AC/DC reaches a saturation point, whereas in all other setups model performance continues to improve with training budget.

### 4.2.2 Sparsity-vs-Accuracy Results

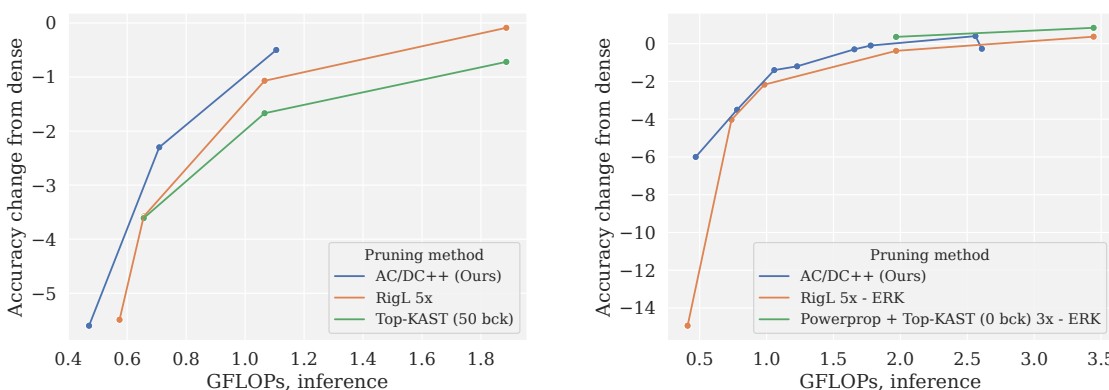

Figure 3: Comparison of Accuracy change from dense baseline as a function of Inference FLOPs for leading sparse training methods, under uniform sparsity constraints (**left**) and global sparsity constraints (**right**). Due to a lack of a standard benchmark, global and Erdős–Rényi Kernel (ERK) sparsity constraints were grouped together. Both sparsity schedules of AC/DC++ (with all layers sparsified and with the first and last layer kept dense) are plotted together.

**Goals and Metrics.**   Based on these results, in this section, we aim to improve the best known sparsity-versus-accuracy trade-offs by performing a thorough ablation over sparsities and training length parameters. We compare our results to the highest-performing previously published sparse training methods. In particular,

| Method | Top-1 accuracy (%) | | Δ Accuracy | Sparsity | Remaining | Inference FLOPs |
| | Dense ($D$) | Sparse ($S$) | $100 \times \frac{(S-D)}{D}$ | (%) | # of params | prop. of dense |
| --- | --- | --- | --- | --- | --- | --- |
| Sparse Training | | | | | | |
| AC/DC (Peste et al., 2021) | 76.8 | 75.03 | -1.77 | 90 | 2.56 M | 0.18 |
| GraNet($s_0 = 0.5$) (Liu et al.) | 76.80 | 74.5 | -1.3 | 90 | - | 0.20 |
| Powerpropagation + Top-KAST FLD (Schwarz et al., 2021) | 76.8 | 75.23 | -1.57 | 90 | - | - |
| Powerpropagation + Top-KAST ERK (Schwarz et al., 2021) | 76.80 | 75.74 | -1.06 | 90 | - | 0.24 |
| RIGL ERK 1x (Evci et al., 2020) | 76.80 | 73.00 | -4.94 | 90 | - | 0.24 |
| RIGL-ITOP ERK 1x (Liu et al., 2021) | 76.80 | 73.82 | -2.98 | 90 | - | 0.24 |
| ST-3 (Vanderschueren & Vleeschouwer, 2023) | 77.10 | 75.28 | -1.82 | 90 | - | 0.24 |
| STR (Kusupati et al., 2020) | 77.01 | 74.31 | -3.51 | 90.23 | 2.49 M | - |
| Variational Dropout (Molchanov et al., 2017) | 76.69 | 73.84 | -3.72 | 90.27 | 2.49 M | - |
| Post-training sparsification | | | | | | |
| Global Magnitude (Singh & Alistarh, 2020) | 77.01 | 75.15 | -2.42 | 90 | 2.56 M | - |
| WoodFisher (Singh & Alistarh, 2020) | 77.01 | 75.21 | -2.34 | 90 | 2.56 M | - |
| Extended sparse training | | | | | | |
| AC/DC++ 5x (this work) | 78.78 | 78.49 | -0.29 | 90 | 2.60 M | 0.2 |
| AC/DC++ FLD 5x (this work) | 78.78 | 78.6 | -0.18 | 90 | 4.45 M | 0.22 |
| GMP FLD 1.5x (Gale et al., 2019) | 76.69 | 75.16 | -1.53 | 90 | - | - |
| GraNet($s_0 = 0.5$) 2.5x (Liu et al.) | 76.80 | 76.4 | -0.4 | 90 | - | 0.20 |
| Powerpropagation+Top-KAST ERK 3x(Schwarz et al., 2021) | 76.80 | 77.16 | +0.36 | 90 | - | 0.24 |
| RIGL ERK 5x (Evci et al., 2020) | 76.80 | 76.42 | -0.38 | 90 | - | 0.24 |
| RIGL-ITOP ERK 5x (Liu et al., 2021) | 76.80 | 75.50 | -1.30 | 90 | - | 0.24 |
| Sparse Training | | | | | | |
| AC/DC (Peste et al., 2021) | 76.8 | 73.14 | -3.66 | 95 | 1.28 M | 0.11 |
| GraNet($s_0 = 0.5$) (Liu et al.) | 76.80 | 72.3 | -6.5 | 95 | - | 0.12 |
| Powerpropagation + Top-KAST FLD(Schwarz et al., 2021) | 76.8 | 73.25 | -3.55 | 95 | - | - |
| RIGL ERK 1x (Evci et al., 2020) | 76.80 | 70.00 | -8.85 | 95 | - | 0.12 |
| ST-3 (Vanderschueren & Vleeschouwer, 2023) | 77.10 | 74.46 | -2.64 | 95 | - | 0.13 |
| STR (Kusupati et al., 2020) | 77.01 | 70.40 | -8.58 | 95.03 | 1.27 M | - |
| Variational Dropout (Molchanov et al., 2017) | 76.69 | 71.81 | -6.36 | 94.94 | 1.30 M | - |
| Post-training sparsification | | | | | | |
| Global Magnitude (Singh & Alistarh, 2020) | 77.01 | 71.72 | -6.29 | 95 | 1.28 M | - |
| WoodFisher (Singh & Alistarh, 2020) | 77.01 | 72.12 | -6.89 | 95 | 1.28 M | - |
| M-FAC (Frantar et al., 2021) | 77.01 | 72.6 | -4.41 | 95 | 1.28 M | - |
| Extended sparse training | | | | | | |
| AC/DC++ 10x (this work) | 78.78 | 77.27 | -1.48 | 95 | 1.33 M | 0.13 |
| AC/DC++ FLD 10x (this work) | 78.78 | 77.7 | -1.08 | 95 | 3.28 M | 0.14 |
| GMP FLD 1.5x (Gale et al., 2019) | 76.69 | 72.71 | -3.98 | 95 | 1.28 M | - |
| RIGL ERK 5x (Evci et al., 2020) | 76.80 | 74.63 | -2.17 | 95 | 1.28 M | 0.12 |
| Sparse training | | | | | | |
| AC/DC (Peste et al., 2021) | 76.8 | 68.44 | -9.36 | 98 | 0.7 M | 0.06 |
| ST-3 (Vanderschueren & Vleeschouwer, 2023) | 77.10 | 70.46 | -6.64 | 98 | - | 0.07 |
| STR (Kusupati et al., 2020) | 77.01 | 70.40 | -8.58 | 98 | - | - |
| Variational Dropout (Molchanov et al., 2017) | 76.69 | 64.52 | -15.87 | 98.57 | 0.36 M | - |
| Post-training sparsification | | | | | | |
| M-FAC (Frantar et al., 2021) | 77.01 | 67.5 | -9.51 | 98 | - | - |
| WoodFisher (Singh & Alistarh, 2020) | 77.01 | 65.55 | -11.46 | 98 | 0.51M | - |
| Extended sparse training | | | | | | |
| AC/DC++ 10x (this work) | 78.78 | 74.06 | -4.72 | 98 | 0.51 M | - |
| AC/DC++ FLD 10x (this work) | 78.78 | 76.6 | -2.28 | 98 | 2.58 M | 0.09 |
| Sparse training | | | | | | |
| ST-3 (Vanderschueren & Vleeschouwer, 2023) | 77.10 | 63.88 | -13.22 | 99 | - | 0.04 |
| Extended sparse training | | | | | | |
| AC/DC++ FLD 10x (this work) | 78.78 | 72.7 | -6.08 | 99 | 2.34 M | 0.06 |
| RIGL ERK 5x (Evci et al., 2020) | 76.80 | 61.86 | -15.94 | 99 | - | 0.05 |
| RIGL ERK 10x (Evci et al., 2020) | 76.80 | 63.89 | -12.91 | 99 | - | 0.05 |
| RIGL ERK 50x (Evci et al., 2020) | 76.80 | 66.94 | -9.86 | 99 | - | 0.05 |
| RIGL ERK 100x (Evci et al., 2020) | 76.80 | 68.15 | -8.65 | 99 | - | 0.05 |

Table 1: Comparison between modern sparse training methods on ImageNet-1k with ResNet-50 models for various sparsity targets. ERK refers to the Erdos-Renyi Kernel sparsity distribution. FLD refers to the first and last layers being dense (AC/DC++) or the first layer being dense and the last layer being 80% sparse (GMP, PowerPropagation).

we compare an extended-training version of AC/DC, which we call AC/DC++, results reported in the original RigL, ST-3, and Powerpropagation papers, as well as many other existing pruning methods.[4]. All methods are described in Section 3. In cases where the authors conducted extended training using their method, we present those numbers, and we use the FLOPs-optimized ST-3$^\sigma$ variant. AC/DC++ candidate models were trained for four preset training lengths (1x, 2.5x, 5x and 10x the standard ImageNet training time on ResNet50) at all sparsity levels, and we chose the best results obtained by ablating over the length of the training run.

---

[4]The most successful Powerpropagation approach presented in the paper combines this method with Top-KAST; we use this benchmark, as it performs better than Top-KAST alone.

As different methods have different computational budgets and different dense baselines, to ensure a fair comparison, we examine the model performance both in terms of *Top-1 Validation accuracy*, and the *Top-1 Validation accuracy difference from the corresponding dense baseline*. We use the best available numbers originally reported in the papers introducing the methods for comparisons.

**Experimental Setup.** We compare two pruning regimes. First, we consider *Uniform Pruning*, in which every layer is pruned exactly to the target sparsity, except for the first and last layer, which are left dense. Second, we consider the *Global/Nonuniform Pruning* regime, in which the sparsity budget is set globally. Different works apportion the global budget differently, and also differ with respect to which parts of the network are subject to the global constraint. In particular, Extended GMP (Gale et al., 2019) and Top-KAST do not prune the first layer, prune the last layer to a fixed 80% sparsity, and prune the other layers using a global magnitude criterion. RigL uses an Erdős–Rényi-Kernel distribution for layer sparsity targets, and leaves only the first layer dense. The original AC/DC work uses global sparsity and prunes all convolutional and FC layers. Therefore, to create a more fair comparison, we consider estimated Floating-Point Operations (FLOPs) necessary for inference; these are computed as in (Evci et al., 2020). Using FLOPs also equalizes methods across slight variations in ResNet50 architectures, and so we use it also for the Uniform pruning comparison. In addition, we use two pruning schedules for AC/DC++: one which leaves the first and last layer dense and prunes the remaining layers using a global magnitude criterion, and one that prunes all layers using the global magnitude criterion. We do not ablate between the two, but rather present both sets of results in Figure 3 (jointly) and Table 1 (separately).

We emphasize two key points regarding our comparisons:

1. Looking at accuracy alone favors AC/DC++, as it has a higher dense baseline: since we use several recent training innovations, the dense model can reach 78.78% dense accuracy over 100 epochs. Therefore, it becomes more challenging to maintain the performance of the dense model for highly sparse model compared to less-optimized baseline.

2. This is why we also examine *accuracy difference relative to the dense baseline*: this favors other methods, as they are benchmarked against a standard-recipe model that reaches lower 76.8% accuracy (77.1% for ST-3).

**Results.** The results are presented in Figure 3 and Table 1. We observe that, for uniform pruning budgets, the AC/DC++ models outperform other methods, both in terms of absolute and relative validation accuracy. This is true even when we consider extended-training schedules for other methods, although we believe we are the first to systematically investigate the impact of increasing training schedules at these sparsity levels.[5] When looking at models trained with global pruning budgets, we observe that AC/DC++ obtains the highest absolute validation accuracy, compared to results reported previously in literature. When considering accuracy change from the dense line, AC-DC++ loses less accuracy than other methods at very high sparsities (lowest FLOPs), despite having the highest-performing dense baseline; at lower sparsity (90%), it is competitive with other extended training methods.

### 4.3 Additional validations and ablations

We performed additional analysis and ablations, to validate the performance and hyperparameters of the AC/DC++ model, and better understand the factors contributing to its high performance. These studies are summarized briefly below, and available in the Appendix.

#### 4.3.1 Additional Evaluations

**Additional quality evaluations.** Having demonstrated that extended training has a strong positive effect on sparse model top-1 test accuracy, we further investigate the impact of extended training on other aspects of model quality. We consider two additional quality metrics: their performance in transfer learning scenarios and robustness to common image perturbations.

---

[5]In prior work, RigL executed >5x extended training for a 99%-sparse model only (Evci et al., 2020).

Iofinova et al. (2022) demonstrated that equally sparse models with comparable performance on the original task can vary widely in their performance once finetuned on other, smaller transfer tasks. We compare the transfer performance of dense and 95% sparse AC/DC++ models, both trained for 100 and 1000 epochs in two transfer learning regimes: linear finetuning, where the hidden layers of the model are trained only on the larger (ImageNet) task, and only the final FC layer is trained on the transfer task, and full-network finetuning, where all layers are finetuned on the transfer task. We find that extended training improves the transfer performance for both transfer scenarios for 95% sparse models, but is largely neutral for dense models. Full details of the experiment and evaluation are given in Appendix E.

We test robustness by measuring model performance on the ImageNet-C dataset (Hendrycks & Dietterich, 2019), which digitally adds 19 types of perturbations to the ImageNet-1K validation set. (Liebenwein et al., 2021) and (Hooker et al., 2019) have found that compressed models are less robust under many types of perturbations, compared to dense models. As before, we consider dense and 95% sparse AC/DC++ models trained for 100–1000 total epochs. We find that robustness to perturbations increases with training time for sparse models, but stays the same for dense ones. Full details of the experiment and evaluation are given in Appendix F.

**Additional models.**   In Appendix D,we show that extended training with AC/DC++ produces state-of-the-art results on the MobileNet-V1 architecture as well.

### 4.3.2   Parameter Ablations

**AC/DC dense fraction.**   We note that AC/DC is relatively expensive in terms of training FLOPs as compared to other sparse training methods. In Appendices H and L we investigate if this can be improved by shortening the duration of the decompression phase relative to the compression phase, and by using a lower, but nonzero sparsity during the decompression phases of AC/DC, respectively. We find that, for models with a target sparsity of 95%, spending 50% of the training time in each phase is optimal, consistent with the original model recipe. However, shortening the decompression phase so that the model spends only 20% of the training time in that phase has a small 0.2% accuracy drop. Additionally, we find that, for 95% sparse models, decompression phases that are up to 70% sparse have matching or better performance to the original AC/DC models with 0% sparse decompression phases. Therefore, if minimizing floating-point operations during training is an objective, it is possible to make the training more efficient in that regard.

**AC/DC phase duration.**   In Appendix I we confirm that for ResNet50 models trained on ImageNet and assuming equally-sized compression and decompression phases, the 5-epoch phase duration used in the initial paper is optimal.

**Impact of weight decay.**   in Appendix L, we consider the impact of weight decay on AC/DC model performance and sparsity. We find that using high values of weight decay (1e-3) results in models that stay largely sparse even during the decompression phase, and that are considerably less accurate. Conversely, using low values of weight decay (1e-5 and 1e-6) result in models with very low sparsity during the decompression phase, and also decreased performance.

### 4.3.3   Additional Analyses

**Mask Analysis.**   We investigate the importance of mask updates during sparse training. Specifically, we examine how much masks change as training progresses. We find (see Figure 4 (left)) that for both RigL and AC/DC, masks change substantially more early in the training, with an Intersection/Union scores of 0.3-0.4 for AC/DC 95% sparse models and 0.92-0.95 for RigL during the first 20% of training steps (recall that RigL masks are updated far more frequently than AC/DC masks). Later in the training, masks stabilize, with less change in consecutive masks. Full results are provided in Appendix J,

**Loss landscape analysis.**   In Appendix M, we investigate the sharpness of the loss landscape (as measured by an approximation to the highest eigenvalue of the Hessian matrix at the point of convergence). We find (see Figure 4 (left)) that, across all methods, sharpness increases with the length of the training run, indicating

that sharper minima require extended training to be reached via SGD. Additionally, sharpness decreases with the increase of sparsity. All sparse training methods attain lower sharpness compared to the dense model.

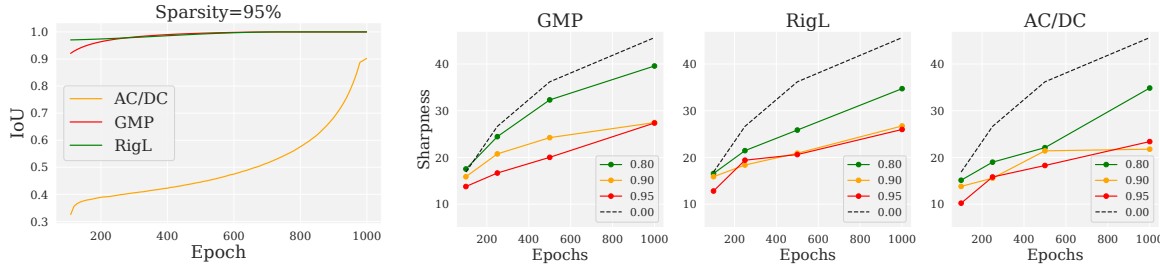

Figure 4: (**left**) Mask IoU between two consecutive checkpoints at 95% sparsity. (**right**) Sharpness (highest eigenvalue) of the loss surface vs number of epochs. Dashed lines correspond to the dense model.

**Structured sparsity.** In Appendix K, we measure the number of channels that have been completely zeroed out during (unstructured) sparse training. We find that AC/DC and AC/DC++ models have a considerable amount of sparse channels, with AC/DC++ models having considerably higher structured sparsity than AC/DC for every sparsity.

**Comparison with smaller dense model.** In Appendix G we confirm that a 95% sparse ResNet50 model substantially outperforms a half-width dense ResNet50 model trained for the same budget.

**Different sparsity patterns.** In Appendix N, we confirm that adding constraints on the sparsity pattern (such as uniform per-layer sparsity and block-4 sparsity) lowers model accuracy.

## 5 The Difficulty of Sparse Transfer in Language Modelling

Next, we extend the analysis to language models, specifically to the very common scenario in which a large language model (BERT-base) is adapted to a specific task via *finetuning*. Thus, here we will examine the impact of sparsity target, number of iterations, loss function, and hyper-parametrization on the optimal recipe for the task of *finetuning* sparse models on the downstream dataset.

In the context of our study, this setup naturally leads to the following questions: *"do finetuned sparse language models suffer from being undertrained on the downstream task?"*, and *"if yes, does the simple recipe of extended training suffice to mitigate the issue?"*. In this section, we will show that when dense finetuning recipes are used for sparse transfer learning in language models, the resulting models are indeed undertrained and have poor transfer performance. However, we also note an additional difficulty: extended training does not suffice to mitigate the issue, because sparse language models quickly shift from being undertrained to an overfitting regime. The latter is a far larger problem in language understanding tasks than in visual ones, which is likely why we don't observe the same issues with visual transfer learning in Appendix E - there we simply use a long finetuning schedule in all cases. In this section, we explore the problem of balancing under- and over-training in sparse language models and propose a sparse finetuning recipe for creating properly tuned sparse models.

### 5.1 Under Standard Dense Transfer Learning Recipes, Sparse Models are Undertrained

**Experimental Setup.** In our experiments, we make use of open-sourced *sparse* pre-trained BERT-base models obtained by (Kurtic et al., 2022). On top of these, we apply various transfer learning recipes to obtain fine-tuned sparse models on datasets from the popular GLUE benchmark (Wang et al., 2018). For fair comparisons with results from prior work, we employ early stopping for all methods. We provide more details about each dataset in Appendix O.

The most popular and widely adopted dense transfer learning recipe consists of fine-tuning all weights with linearly decaying learning rate for as much as two or three epochs on the target downstream task. In Table 2 we present results obtained with this approach when applied to sparse models, and denote it as a *dense-transfer recipe*. Under the same transfer learning recipe, we clearly observe significant gaps (up to 14 accuracy points on RTE and CoLA) between the transfer accuracy of the *dense model* (*Dense BERT-base*), and the transfer accuracy of the *sparse model* (*Dense-transfer recipe*).

## 5.2   Extended Training Shifts from Undertraining to Overfitting

Observing that the dense transfer learning recipe does not produce competitive sparse finetuned models, we attempt to scale the length of the recipe to mitigate undertraining. Surprisingly, for sparse language models, this simple technique does not yield a unique setup with consistently better results as models quickly shift from undertraining to an overfitting regime, in which training loss goes to zero, while validation accuracy decreases sharply. To demonstrate this overfitting effect with the extended recipe, in Table 2, we compare results obtained with this approach (*Extended dense-transfer recipe*) against doing a full sweep of finetuning runs with rescaled recipes to #epochs $\in \{1, 2, 3, ..., \text{extended} - 1\}$ (*Full sweep of rescaled recipes*).

Table 2: Sparse-transfer performance of 90% sparse pre-trained BERT-base model on the dev-set of the corresponding GLUE task, obtained with dense and extended dense (#epochs=8) transfer learning recipes, as well as with the full sweep of rescaled recipes (#epochs $\in \{1, 2, ..., 7\}$).

| Sparse-transfer | RTE Acc | QNLI Acc | MRPC Acc | SST-2 Acc | CoLA Mcc | STS-B Pear | MNLI Acc | QQP Acc |
|---|---|---|---|---|---|---|---|---|
| Dense BERT-base (baseline) | 66.1 | 91.3 | 85.5 | 93.0 | 56.8 | 88.9 | 84.6 | 91.5 |
| Dense-transfer recipe | 52.4 | 88.9 | 82.8 | 91.2 | 42.5 | 87.1 | **82.2** | 90.0 |
| Extended dense-transfer recipe | 55.2 | 88.7 | **85.6** | 91.4 | 47.2 | 87.6 | 81.6 | 90.3 |
| Full sweep of rescaled recipes | **57.0** | **89.3** | 84.1 | **92.0** | **48.5** | **88.0** | 82.2 | **90.4** |
| Best recipe length | 5 ep | 2 ep | 5 ep | 2 ep | 7 ep | 4 ep | 3 ep | 5 ep |

The results suggest that with the existing recipes, there is no one-size-fits-all solution. Versions of this rescaling approach have been utilized by prior works like (Kurtic et al., 2022) and (Zafrir et al., 2021) to obtain accurate sparse models on various downstream datasets. However, this approach comes with a huge computational burden: for each rescaled recipe, a full hyperparameter sweep over relevant parameters has to be done in order to obtain competitive finetuned sparse models. Due to practicality and associated costs, this is not a desirable solution in practice.

## 5.3   Sparse Transfer Learning for Language Models

In the previous section, we have demonstrated the following three problems with the existing approach of either using the dense finetuning recipe, or simply extending it for sparse finetuning:

1. following dense-transfer recipes, sparse language models are undertrained;

2. even at high sparsities, these models can still exhibit overfitting behavior under the extended training regime;

3. finding the optimal recipe to mitigate undertraining and overfitting has major computational burdens.

To address these issues, we propose a simple approach for sparse transfer in NLP, which produces highly accurate and competitive sparse models on a wide range of downstream datasets with minimal hyperparameter tuning. Our technique is inspired by the idea of gradual layer unfreezing presented in the ULMFiT framework (Howard & Ruder, 2018), which introduced a universal framework for fine-tuning *dense* language models for text-classification tasks, with a focus on LSTM models (Hochreiter & Schmidhuber, 1997; Merity et al., 2017). Based on ULMFiT and findings of (Yosinski et al., 2014), which suggests that different layers

Table 3: Our sparse-transfer performance of 90% sparse pre-trained BERT-base model on the dev-set of the corresponding GLUE tasks, benchmarked against the current state-of-the-art sparse-transfer results from Prune OFA (Zafir et al., 2021) and oBERT (Kurtic et al., 2022).

| Sparse-transfer | RTE Acc | QNLI Acc | MRPC F1 / Acc | SST-2 Acc | CoLA Mcc | STS-B Pear / Spear | MNLI m / mm | QQP Acc / F1 |
|---|---|---|---|---|---|---|---|---|
| Dense BERT-base | 66.1 | 91.3 | 89.8 / 85.5 | 93.0 | 56.8 | 88.9 / 88.5 | 84.6 / 83.4 | 91.5 / 88.5 |
| Prune OFA (Zafir et al., 2021) | N/A | 89.1 | N/A | 90.9 | N/A | N/A | 81.5 / 82.4 | **90.9** / **87.6** |
| oBERT (Kurtic et al., 2022) | 57.0 | 89.3 | 89.3 / **85.6** | **92.0** | 48.5 | **88.0** / **87.6** | 82.2 / 82.5 | 90.4 / 87.1 |
| This work | **60.1** | **90.5** | **89.7** / 85.2 | 91.8 | **51.4** | 87.2 / 87.1 | **83.7** / **83.8** | **90.9** / **87.6** |

capture different information and therefore should be fine-tuned to different extents, we adopt the idea of gradual unfreezing and adjust it for *transformer-based (Vaswani et al., 2017) sparse* language models.

More specifically, we focus on the popular BERT-base model which consists of three groups of layers: embeddings, 12 identical transformer blocks, and a task-specific classifier head. Sparsified versions of this model, which are the main interest of this work, prune all linear layers across all transformer blocks, which is the standard practice in literature (Sanh et al., 2020; Kurtic & Alistarh, 2022; Kurtic et al., 2022; Zafir et al., 2021) and brings the best accuracy-vs-latency trade-offs (Kurtic et al., 2022).

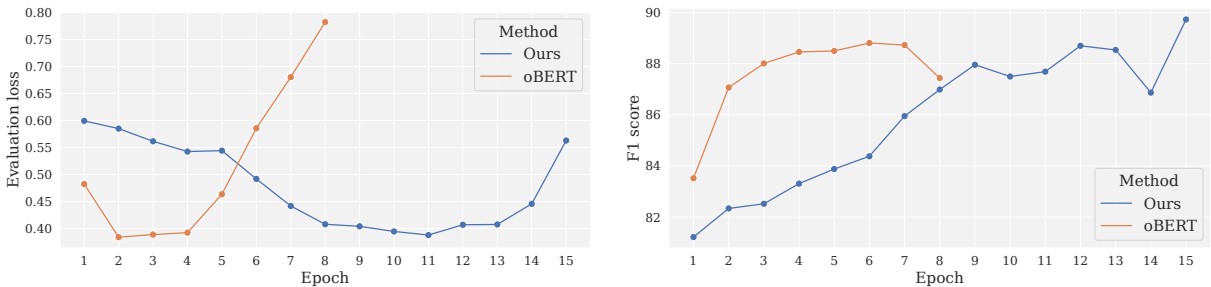

Figure 5: Evaluation loss (lower is better) and F1 score (higher is better) during sparse-transfer with oBERT (Kurtic et al., 2022) and our approach on MRPC dataset.

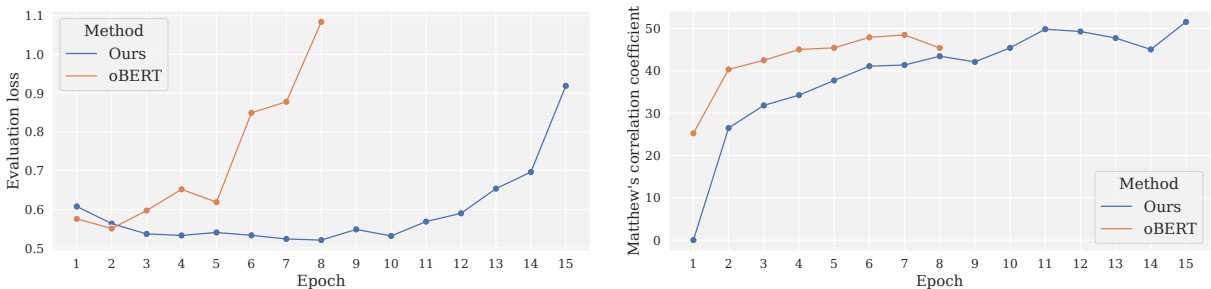

Figure 6: Evaluation loss (lower is better) and Matthew's correlation coefficient (higher is better) during sparse-transfer with oBERT (Kurtic et al., 2022) and our approach on CoLA dataset.

Our approach can be summarized as follows. For each downstream task, we start from a sparse pre-trained model produced by (Kurtic et al., 2022) and randomly initialize a task-specific classifier head. Then we freeze all embeddings and sparsified linear weights, while keeping their biases and corresponding LayerNorm (Ba et al., 2016) layers unfrozen and trainable. We start by finetuning only the classifier head and all other trainable parameters (biases and LayerNorms) for one epoch, and then follow the same process from back-to-front by unfreezing the unpruned linear weights in preceding transformer blocks. After the last layer is unfrozen and finetuned, we continue finetuning all layers together for one more epoch.

Given that at each epoch we have a different model architecture (one more sparse transformer block unfrozen relative to the previous epoch), we finetune it with the linearly decaying learning rate and then rewind back to the initial value for the next epoch. We have also tried the slanted triangular learning rate schedule proposed in ULMFiT, but we found the warmup phase not very helpful as it is known that sparse language models usually require much higher learning rates relative to their dense counterparts in order to train and converge successfully (Kurtic & Alistarh, 2022).

To validate the effectiveness of our proposed sparse transfer approach, we benchmark it against the two current state-of-the-art sparse-transfer results presented in *Prune Once for All (Prune OFA)* (Zafrir et al., 2021) and *The Optimal BERT Surgeon (oBERT)* (Kurtic et al., 2022) papers. The former makes use of knowledge distillation from a finetuned dense teacher model, while the latter uses a full sweep over extended and rescaled dense transfer recipes, such as the ones we presented in Section 5.2. As can be seen from Table 3, our approach outperforms highly competitive results by Prune OFA in all, and oBERT in eight out of twelve datasets, setting new state-of-the-art accuracy-vs-sparsity results for many tasks in the GLUE benchmark suite. It is worth emphasizing that all of our results are obtained with significantly less hyperparameter tuning than the other two competing methods, which aligns with our goal of finding a stable one-size-fits-all solution for the sparse-transfer problem. We search and tune the initial learning rate in {1e-4, 2e-4, 3e-4}, and dropout in {0.05, 0.1}, and report mean performance over the two best runs. Thus, our grid consists of only 6 different combinations for each considered dataset, whereas competing approaches sweep over 54 (Zafrir et al., 2021) and 24 (Kurtic et al., 2022) different combinations. It is worth emphasizing that all of the considered methods, including ours, have noticeable variability in results on small datasets across different seeds and hyperparameter configurations, which aligns with findings of (Devlin et al., 2019).

To better understand what happens during our proposed sparse transfer learning setup, and to develop an intuition about why it is able to provide stable and competitive results across many different datasets ranging in sizes from 2.4k (RTE) and 392k (MNLI) labeled samples, we visualize evaluation loss and evaluation accuracy metrics over the entire transfer learning process in Figures 5 and 6. As can be seen, our approach enables slower and therefore more stable transfer learning on the target datasets which effectively prevents overfitting, even though the total number of epochs is two times larger than the extended dense-transfer recipes analyzed in Section 5.2. This aligns with findings in ULMFiT, which demonstrates that gradual unfreezing in combination with a carefully designed learning rate schedule prevents catastrophic forgetting and enables robust transfer learning across a wide range of different downstream tasks.

## 6 Conclusion

In this work, we examined the impact of high sparsity on model training under standard computer vision and natural language recognition scenarios, and provided evidence that traditional training recipes used for dense models are generally too short for sparse training. Starting from this observation, we were able to produce state-of-the-art models for sparse computer vision on two classic benchmarks for pruning: the ResNet50/ImageNet from-scratch training benchmark, and transfer learning from BERT-base on several NLP datasets. Our work focused on the differences between sparse and dense training dynamics and their effect on optimal training, providing additional analysis towards the difficulty of sparse training. In our work we showed that very high levels of both sparsity and accuracy are possible simply by carefully adapting the number of training epochs and using sensible values for basic hyperparameters. We hope that these new results will encourage additional research on adapting training schedules, hyperparameters, optimizers, and data selection that will allow for the creation of sparsely-trained models that match these accuracy targets within a smaller training budget. We leave this as a challenge to the community.

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
