# Part I

# Appendix

# Table of Contents

## A   Training hyperparameters

We adopt and scale the training recipes proposed in the FFCV library (Leclerc et al., 2022). All runs use a linear decaying learning rate with a peak value $\eta_{\max} = 0.5$ at 2nd epoch that is gradually decreased to 0 at the end of the training. We train the models with batch size of 1024.

We use progressive resizing with training on random resized crops (RRC) of size $160 \times 160$ for the first 80% of training and the remaining 20% of training with crops of size $192 \times 192$. We run classifier on $224 \times 224$ center crops for the image resized to 256 px on validation following the standard ImageNet evaluation protocol. Differently from the FFCV recipes, we do not use test-time augmentation with averaging of the prediction on the original image and its flipped version since it requires 2x more inference FLOPs. For MobileNet experiments we adopted smaller value of weight decay 3e-5 following the (Kusupati et al., 2020) work and learning rate $\eta = 1.024$.

Table A.1: Augmentation and regularization procedure used in the work.

| Method | Value |
|---|---|
| Weight decay | 1e-4 |
| Label smoothing $\varepsilon$ | 0.1 |
| Dropout | ✗ |
| H.flip | ✓ |
| RRC | ✓ |
| Blurpool | ✓ |
| Progressive resizing | ✓ |
| Test crop ratio | 0.875 |

## B  Sparse training algorithm hyperparameters

All sparse training methods use the same optimizer and learning rate scheduler hyperparameters and differ only in the specifics of weight pruning / regrowth procedure. In our work we prune only weights of convolutional (we do not prune biases and batch norm parameters) and keep the first convolution as well as the classification head (the last linear layer outputting the logits for classes) dense. This setup is common in the literature.

**AC/DC**  In our AC/DC training setup we train the model without sparsity for the first 10% of the training and then alternate between sparse and dense training each 5 steps following the original paper (Peste et al., 2021). We run the last compression step and last decompression step for 10 and 15 epochs respectively, again following the prescription from the paper. We have ablated the duration of the compression and decompression phases and 5 epochs appeared to yield the best performance.

**RigL**  In the original paper (Evci et al., 2020) authors train the models with fixed target sparsity and periodically update the fraction of connections in each pruned layer $l$ following the cosine decay rule:

$$\frac{\alpha}{2}\left(1 + \cos\left(\frac{\pi t}{T}\right)\right)(1 - s_l) \tag{3}$$

Above $s_l$ is the sparsity of a given layer $l$ and $\alpha$ is the initial fraction of updated connections. The choice of update frequency $\Delta T$ has a strong impact on the method performance and in the original work authors obtained the best performance with $\Delta T \simeq 100 - 300$ steps, which corresponds roughly to 0.4-1.2 epochs on ImageNet with the batch size used in the work. In our work we updated connections every epoch to have setup close to the original work. We set $\alpha = 0.3$ as in the one used in the RigL work. Following the original work, we train with a fixed sparsity mask for the last 25% of training. For global sparsity we apply the ERK (Erdős–Rényi-Kernel) sparsity profile as in the original paper as it was shown to produce the best performance for a given sparsity.

**GMP**  We gradually increase the sparsity from 0% (dense model) to the target sparsity following a cubic interpolation law. Sparsity is increased every 5 epochs. We train with a fixed mask for the last quarter of training duration as in RigL.

## C  Extended undertraining analysis

In Section 4.1, we examined the connection between sparsity, training loss, and validation accuracy in ResNet-50 models trained on Imagenet. Here, we present an alternative metric to the training loss, in the form of Entropy on the *validation* set, as well as validate our results on the CelebA dataset.

## C.1    The validation entropy metric

Low prediction entropy implies that the prediction weight is largely concentrated in a single class, while a high entropy suggests that it is spread out over several classes. Intuitively, the entropy of the model is related to its "confidence" in predictions, and is independent of whether the predictions are correct (and so can be measured on data for which labels are not available). Conversely, low training loss measures the model's fit to the training data. The prediction entropy on test or validation data is also closely connected to the model's calibration.

The similarity of the two metrics can be seen directly from their formulas. We compute the cross-entropy loss and prediction entropy by taking the softmax over the vector of output values of the network and then applying the respective standard formulas, where the cross-entropy is taken with respect to the correct label distribution for the model (1 for the correct class and 0 otherwise). For an output of a network outputting a vector $Z = (z_1, z_2, ..., z_C)$ of size $C$ with correct label $L$, the entropy $H$ and the cross-entropy $CE$ are given by the following formulas:

$$H(Z) = -\sum_{i=1}^{C} \frac{e^{z_i}}{\sum_{j=1}^{C} e^{z_j}} \log \left( \frac{e^{z_i}}{\sum_{j=1}^{C} e^{z_j}} \right) \quad \text{and} \quad CE(Z) = -\log \left( \frac{e^{z_L}}{\sum_{j=1}^{C} e^{z_j}} \right). \tag{4}$$

We expect a sufficiently large and well-trained model to have (a) low loss on the training data and (b) fairly low average prediction entropy, while a model that is not well-trained to have high prediction entropy. However, as is conventionally known, continued training on dense and low-sparsity models resulting in overfitting will lower these metrics further.

The experimental setup is exactly as in Section 4.1, just with the additional computation of the Entropy metric on the validation set. We present the results in Figure C.1, which also includes the plots from Figure 1 for ease of comparison.

We observe that, for all sparsity levels, the Train Loss and Validation Entropy metrics behave very similarly. Most crucially, we observe that, just as with the training loss, there is an 'optimal' entropy level across model sparsities, with models whose validation entropies fall below that value having lower validation accuracy. We believe this provides additional evidence that these metrics may be used to detect overfitting in models and to find the optimal training duration.

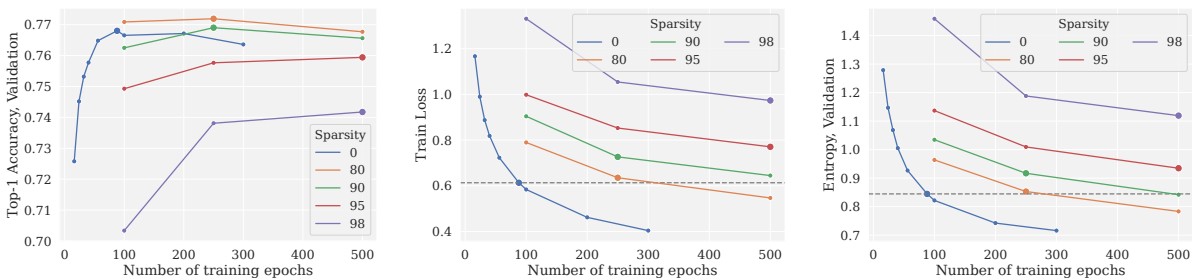

Figure C.1: Average validation accuracy (left), Train loss at final epoch (center), and Entropy (right) for sparse and dense ImageNet models trained for different numbers of epochs. The highest-accuracy model for each sparsity level is highlighted with a larger marker. The cross-entropy loss and entropy level of the dense model is also shown with a dashed line, to simplify comparison.

## C.2    CelebA results

To validate our findings, we repeat this experiment on the Celeb-A Dataset (Liu et al., 2015). This dataset consists of a combined 202599 face images of 10177 celebrities collected from the public domain, automatically

cropped to the face, and annotated with 40 binary labels. Due to its content, this dataset is frequently used to study bias in machine learning models, and has also been used in studies on the effect of sparsity on bias (Hooker et al., 2019; Iofinova et al., 2023).

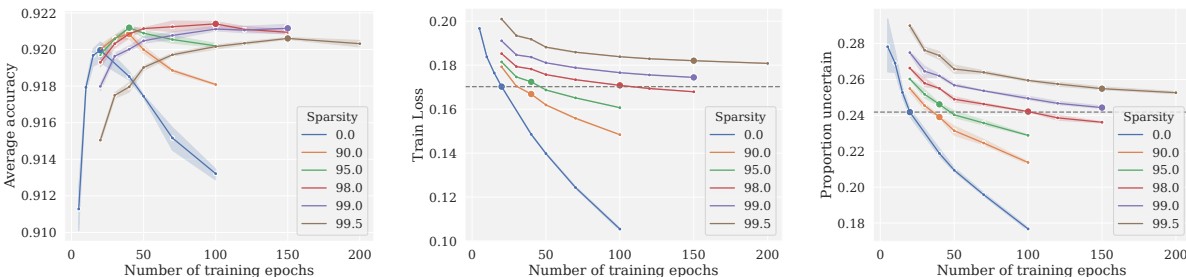

Figure C.2: Average validation accuracy (left), train loss at final epoch (center), and uncertainty (right) for sparse and dense CelebA models trained for different numbers of epochs. The highest-accuracy model for each sparsity is highlighted with a larger marker. The cross-entropy loss and entropy level of the dense model is also shown with a dashed line, to simplify comparison.

**Experimental Setup.**    Following (Iofinova et al., 2023), we train the ResNet18 architecture on this task. We train dense and sparse (90%, 95%, 98%, 99%, and 99.5%) models for a varying number of epochs (5-200); in some cases the very low or very high-epoch runs are skipped if it is clear that the duration will not be optimal for that sparsity. Sparse models are produced with a variant of AC/DC, in which the sparsity of the sparse phases ramps up progressively from 90% to the final target sparsity; this is necessary to prevent layer collapse at very high sparsities. Unlike the ImageNet experiments, here the phase length varies somewhat with duration, due to the extremely short duration of some runs. For each experiment, characterized by a sparsity/epoch-length pair, we measure the accuracy and training loss of the resulting model. In addition, following (Iofinova et al., 2023), we measure the *uncertainty*, rather than entropy, of the model predictions on the test set. The prediction uncertainty is computed as follows: first, the sigmoid operator is applied to each logit's output in order to obtain a pseudo-probability of a positive label between 0 and 1; if this quantity is between 0.1 and 0.9, the prediction is considered uncertain. We then compute the proportion of uncertain predictions across the validation dataset.

**Results.**    The results are presented in Figure C.2. We observe that, consistent with our earlier observations on ImageNet, the optimal training duration goes up with sparsity, with dense models reaching their optimal accuracy at 25 epochs, and 99% and 95% sparse models at 150 epochs. Further, we observe that, even as training loss and test uncertainty always decrease with longer training, the overall training loss and proportion of uncertain predictions goes up with sparsity at a fixed training length. As in the ImageNet example, the highest-performing models at each sparsity have a similar training loss of about 0.17 and mean prediction uncertainty of about 24%, except for the very sparse 99.5% model, which has slightly higher 26% uncertain predictions.

**Discussion.**    We interpret these results as corroborating evidence that sparse models require longer training compared to dense models to achieve optimal accuracy before the overfitting starts to take place.

## D   MobileNet results

In addition, we conducted sparsification of the MobileNet-V1 model (Howard et al., 2017), a CNN optimized for inference on mobile devices. We applied AC/DC for 1000 epochs with sparsity targets 75% and 90% using a similar training recipe to ResNet50 except for some differences specified in Appendix A. To achieve the best results we do not prune the input convolution, as well as the classification head and depthwise convolutions, due to their minor contribution to the overall amount of FLOPs and significant impact on the performance of the model.

| Method | Top-1 accuracy (%) | | Relative Drop | Sparsity |
| | Dense ($D$) | Pruned ($P$) | $100 \times \frac{(P-D)}{D}$ | |
| --- | --- | --- | --- | --- |
| AC/DC++ | 72.74 | 72.49 | -0.25 | 75.00 |
| | 72.74 | 70.80 | -1.94 | 90.00 |

Table D.2: Sparse training of MobileNet-V1 with AC/DC++ on ImageNet-1k.

With a longer training recipe one can achieve almost negligible accuracy drop at 75% sparsity and moderate performance decrease at 90%. The results can be found in Table D.2.

## E   Impact of extended training on transfer performance

We further validate our results by using sparse and dense ResNet50 models pretrained on ImageNet for transfer learning for other vision recognition tasks. Transfer learning is a common paradigm in which a large dataset is used to set network weights before they are further fine-tuned on the (typically, smaller) dataset of interest; this approach can provide significant gains over direct training on the smaller task from a random initialization. In this section, we refer to the larger dataset/task (in our case, ImageNet) as the *upstream* task, and the smaller dataset/task as the *downstream* task.

There are two common approaches to transfer learning, the choice of which largely depends on the technological capabilities of the system doing the learning task. If a large amount of compute is available, all model weights can be trained on the downstream task, in a process we call *full finetuning*. Otherwise, if compute is limited, all the layers of the deep neural network except for the final classifier are frozen after downstream training and used purely as a feature extractor, and only the final layer (properly resized) is trained on the downstream task, as a linear classifier on the extracted features. We call this process *linear finetuning*.

We apply both transfer learning approaches to a standard set of twelve downstream tasks, which are frequently used as benchmarks for transfer learning performance(Kornblith et al., 2019; Iofinova et al., 2022; Salman et al., 2020). The twelve datasets include six specialized tasks and six general tasks; a full list of downstream tasks is given in Appendix Table E.4.

We use four variants of ResNet50 models pretrained on ImageNet as the upstream model: dense and 95% globally sparse models trained for 100 and 1000 epochs. Otherwise, we follow the training hyperparameters of (Iofinova et al., 2022), and, also following this work, we compute a single metric across all twelve tasks by computing Average Increase in Error for a set of tasks $T$ over a baseline model. This metric is computed as follows: for each of the downstream tasks, we compare the difference in the error ($1-$Top-1 accuracy) when the baseline (100-epoch dense) model is used for transfer learning, versus when another model (either sparse, or extended-training dense) is used. The increase in error is the difference of these two quantities divided by the dense error; the final metric is the average of these relative differences across all downstream datasets. As argued in (Iofinova et al., 2022), normalizing error differences by the dense error allows us to roughly equalize dataset impact when we compute a single metric across different datasets with very different model performance (e.g., nearly 100% accuracy on the Pets dataset, but about 50% accuracy on the Aircraft dataset. Formally, the metric is computed as:

$$AIE = \frac{1}{|T|} \sum_{t \in T} \frac{Err_{Model,t} - Err_{Baseline,t}}{Err_{Baseline,t}} \tag{5}$$

We compute the AIE using the 100-epoch dense model as the baseline model, and present the results in Table E.3. We observe that the effect of extended training on the dense model is neutral at best - there is a small increase in error with extended training in the linear finetuning regime, and no change in the full finetuning regime. Conversely, the 1000 epoch 95% sparse model outperforms the 100-epoch one across both regimes, with a higher error drop in the linear finetuning regime and a smaller error increase in the full finetuning regime.

| Training Duration | Linear - Dense | Linear - 95% Sparse | Full - Dense | Full - 95% Sparse |
|---|---|---|---|---|
| 100ep. | - | -0.123 | - | 0.112 |
| 1000ep. | 0.026 | -0.182 | 0.000 | 0.062 |

Table E.3: Transfer learning for image recognition - Average Increase in Error over the 100-epoch dense model. Lower is better: positive numbers indicate that the model has, on average, more error than the baseline model; negative numbers indicate that the model has, on average, less error.

| Dataset | Number of Classes | Train/Test Examples | Accuracy Metric |
|---|---|---|---|
| SUN397(Xiao et al., 2010) | 397 | 19 850 / 19 850 | Top-1 |
| FGVC Aircraft(Maji et al., 2013) | 100 | 6 667 / 3 333 | Mean Per-Class |
| Birdsnap(Berg et al., 2014) | 500 | 32 677 / 8 171 | Top-1 |
| Caltech-101(Li et al., 2004) | 101 | 3 030 / 5 647 | Mean Per-Class |
| Caltech-256(Griffin et al., 2006) | 257 | 15 420 / 15 187 | Mean Per-Class |
| Stanford Cars(Krause et al., 2013) | 196 | 8 144 / 8 041 | Top-1 |
| CIFAR-10(Krizhevsky et al., 2009) | 10 | 50 000 / 10 000 | Top-1 |
| CIFAR-100(Krizhevsky et al., 2009) | 100 | 50 000 / 10 000 | Top-1 |
| Describable Textures (DTD)(Cimpoi et al., 2014) | 47 | 3 760 / 1 880 | Top-1 |
| Oxford 102 Flowers(Nilsback & Zisserman, 2006) | 102 | 2 040 / 6 149 | Mean Per-Class |
| Food-101(Bossard et al., 2014) | 101 | 75 750 / 25 250 | Top-1 |
| Oxford-IIIT Pets(Parkhi et al., 2012) | 37 | 3 680 / 3 669 | Mean Per-Class |

Table E.4: Datasets used as downstream tasks for transfer learning with computer- vision models.

# F  Evaluation on ImageNet-C

In order to confirm the robustness of our models against perturbations, we evaluate AC/DC and dense models on the ImageNet-C(Hendrycks & Dietterich, 2019) dataset, which consists of the standard ImageNet validation dataset to which perturbations such as noise, blur, photographic effects (such as contrast enhancement) or weather conditions (such as snow or fog) were digitally added. We use the lowest "1" level of perturbation, as we find that the quality already drops considerably over the clean data. Our results are shown in Figure F.4. We observe that, as expected and also as reported in (Liebenwein et al., 2021), dense models outperform sparse ones on corrupted data. Further, and consistently with our other findings, extended training on the dense model does not improve performance on the ImageNet-C dataset; conversely, such improvement can be seen for all 19 perturbation categories for 95% sparse models.

# G  Comparison with the small dense model

One may ask whether one can achieve similar performance with a smaller dense model. We have taken a smaller version of ResNet50 with 2x smaller hidden dimension (denoted ResNet-50x0.5). We trained the model with the same training procedure as the baseline ResNet-50 model for 1000 epochs. One can observe from Table G.5 that even for smaller number of FLOPs sparse model is much better than the dense one.

Table G.5: Small-dense vs sparse model on ImageNet-1k

| Model (%) | Sparsity (%) | FLOPs | Accuracy (%) |
|---|---|---|---|
| ResNet-50x0.5 | 0 | - | 69.8 |
| ResNet-50 | 95 | - | 77.5 |

# H  AC/DC dense fraction ablation

The original AC/DC paper has equal length of compression and decompression phases and one may raise a question, whether it would be better to train the model longer in compressed state or decompressed to

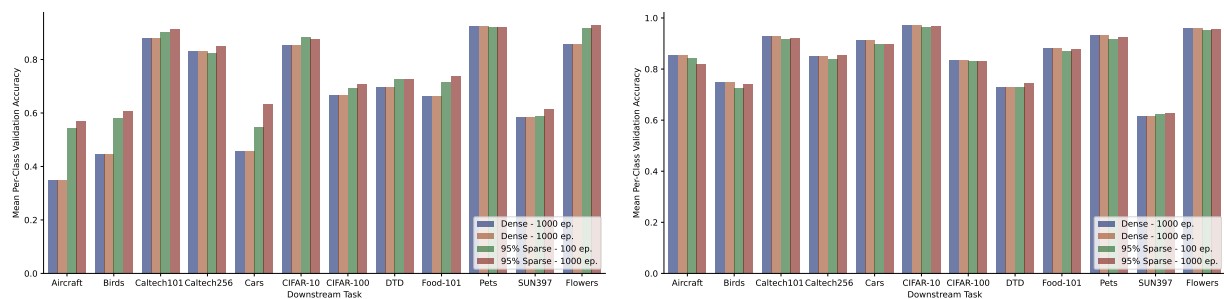

Figure E.3: Linear (left) and Full-network (right) transfer learning mean per-class accuracy for individual datasets.

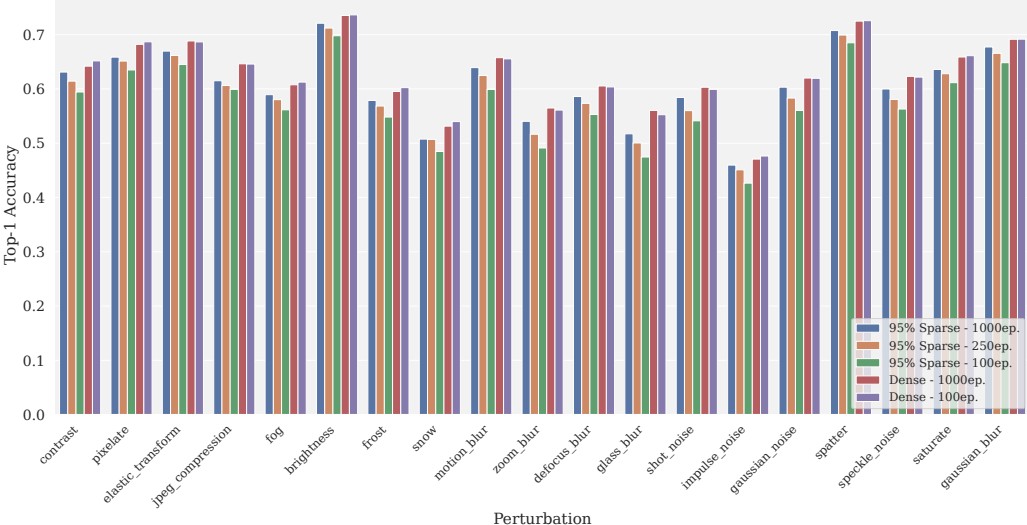

Figure F.4: Linear (left) and Full-network (right) transfer learning mean per-class accuracy for individual datasets.

achieve better performance for a given sparsity target. In the experiments below we train the ResNet-50 model for 1000 epochs with 95% sparsity. It turns out that allocating equal time for dense and sparse training works the best, as one can see from the Table H.6. Here 0 means pruning at some moment of training and finetuning with fixed mask and 1 is the accuracy of dense model. Even when training most of the time in sparse regime one still doesn't lose much in performance.

Table H.6: Ablation on the sparse training fraction.

| Sparse training fraction (%) | Accuracy (%) |
| --- | --- |
| 0 | 79.0 |
| 0.2 | 77.2 |
| 0.5 | 77.5 |
| 0.8 | 77.3 |
| 1 | 75.0 |

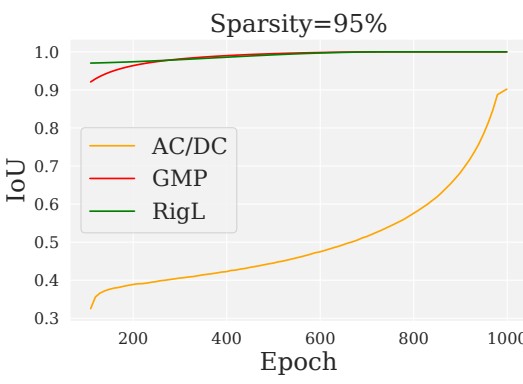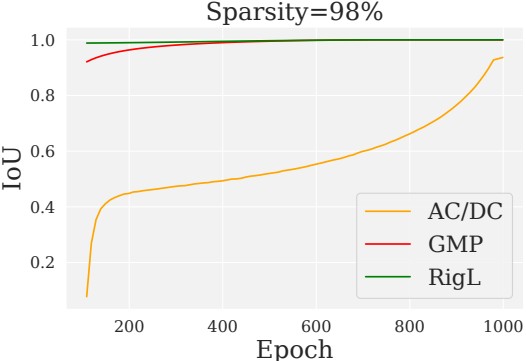

Figure J.5: Mask IoU between two consecutive checkpoints.

# I   AC/DC phase duration ablation

Another hyperparameter that may have a significant impact on performance is the update interval of AC/DC iteration. Intuitively, more frequent updates may allow for more changes in the choice of sparsity of the pruning mask and at the same longer training with fixed mask can be needed to the weights to adapt well for a given sparsity pattern. We have tried different choices of AC/DC iteration duration for ResNet-50 trained for 1000 epochs and 95% global sparsity and the chosen frequency update of 5 epochs is close to the optimum.

Table I.7: AC/DC iteration length ablation.

| Phase length duration (epochs) | Accuracy (%) |
|:---:|:---:|
| 1 | 77.0 |
| 5 | 77.5 |
| 10 | 77.2 |
| 50 | 75.9 |

# J   Mask analysis

Sparse methods considered in our work differ in the amount of sparsity mask exploration. GMP gradually increases sparsity; once the weight is pruned, it is never reintroduced. RigL decreases the fraction of updated parameters following the cosine annealing rule:

$$f_{decay}(t; \alpha; T_{end}) = \frac{\alpha}{2} \left( 1 + \cos \left( \frac{\pi t}{T_{end}} \right) \right) \tag{6}$$

This fraction of connections is dropped and reintroduced in a single step. AC/DC makes all parameters trainable on decompression phases, therefore any parameter could be potentially reintroduced. However, as shown later, some fraction of weights remains zero even on decompression. To measure the difference between two consecutive sparsity masks we compute their IoU (Intersection over Union), the amount of parameters that are nonzero for both checkpoints divided by the amount of parameters that are nonzero in either of checkpoints. High IoU value (close to 1) means that two sparsity masks overlap significantly, whereas low IoU (close to 0) implies low similarly between sparsity masks.

We have taken checkpoints saved on the $109^{th}, 119^{th}, \ldots 999^{th}$ epoch (taken at the end of every AC/DC step and at the same epochs for other methods for a consistent comparison) collected during 1000 epochs runs with 95% and 98% target sparsity and measured IoU between two consecutive masks for parameters being pruned (skipping biases and batch norm parameters). For GMP and RigL, mask IoU can be computed analytically based on the update rule. The evolution of sparsity mask IoU during training is presented on

Figure J.5. One can observe that AC/DC shows significantly stronger mask exploration compared to GMP and RigL. This behavior could account for the better performance of AC/DC as a sparse trainer.

# K  Structured sparsity in unstructured sparse models

| | | Channel sparsity | | | |
|---|---|---|---|---|---|
| Sparsity | Epochs | layer1.0.conv2 | layer2.1.conv2 | layer4.2.conv2 | avg |
| 80 | 100 | 0 | 0 | 0 | 1.07 |
| | 1000 | 18.75 | 0 | 35.35 | 4.91 |
| 90 | 100 | 4.69 | 1.56 | 0.78 | 3.58 |
| | 1000 | 31.25 | 19.53 | 61.91 | 10.68 |
| 95 | 100 | 0 | 7.03 | 22.85 | 8.8 |
| | 1000 | 26.56 | 22.66 | 80.27 | 16.76 |
| 98 | 100 | 7.81 | 12.5 | 75.59 | 23.65 |
| | 1000 | 48.44 | 37.5 | 96.68 | 27.93 |

Table K.8: Fraction of zero output channels for specific layers and on average.

We mainly consider only *unstructured* sparsity, therefore groups of weights and entire channels in particular do not have to be sparse. However, we observed that some of the channels in convolutional kernels are entirely pruned. The effect becomes more pronounced with higher sparsity and longer training.

In Table K.8, we show the fraction of sparse *output* channels for layers in the front, middle and the end of the model and the global fraction of zero channels across all convolutional layers in the model. We observe that channel sparsity increases proportionally with the unstructured sparsity target and with training time, and that, for high sparsity, we obtain a very high proportion of zeroed-out output channels, especially in the wider bottom layers. This is in tune with previous work observing the emergence of structured sparsity in dynamic sparse training (Peste et al., 2021; Iofinova et al., 2022; Yin et al., 2023). We provide a first explanation for this behavior in the next section.

# L  Impact of weight decay on sparsity and model performance

Recall that AC/DC makes all parameters trainable on decompression, therefore, one might expect that sparsity on decompression phase would be near zero. However, we observed that a large fraction of weights remains zero even when the sparsity mask is not imposed. This effect is linked to the channel sparsity discussed in the previous section: once a channel is completely zeroed out, it will continue to receive zero gradients even when the sparsity mask is removed. Further, we provide evidence that this phenomenon is linked to increasing the weight decay mechanism, and in particular the value of this parameter: intuitively, weight decay slowly drives weights towards zero; whenever a full channel is zeroed out in the compression phase, it remains "captured" under the sparsity mask until the end of training.

We investigated this empirically by training ImageNet models on ResNet50 with 95% compression sparsity using AC/DC++ for 100 epochs, varying the weight decay parameter from $10^{-6}$ to $10^{-3}$. Our results are shown in Figure L.6. We observe that the fraction of zero parameters increases with the magnitude of weight decay and also over the course of training. Concretely, we observe that all weight decay values lead to almost fully dense models during the first decompression phase. From there, very low weight decay values of $10^{-6}$ and $10^{-5}$ lead to very little sparsity during the next two decompression phases, and about 10% sparsity during the final five. Conversely, very high weight decay of $10^{-3}$ leads to an immediate jump to 50% sparsity during the second decompression phases, which then increases to 60% over the rest of training. The intermediate value of $10^{-4}$, which is the standard setting and was used in our experiments, leads to an intermediate sparsity, which gradually rises to about 24% over successive decompression phases.

We further present the accuracy of the resulting models in Table L.9. We observe that properly setting the weight decay hyperparameter is crucial for good performance of AC/DC++, and confirm that the standard value $10^{-4}$ is close to the optimal value, as the Table L.9 shows.

**Sparse Decompression.** Building on this observation, we ask how imposing a fixed minimal sparsity also on *decompression stage* impacts the final performance. We conducted a few 100-epoch long AC/DC experiments with 95% target sparsity and set the sparsity on decompression to some fixed value, smaller than compression sparsity. We observe that the performance is almost unaffected up to 80% decompression sparsity, showing that full mask exploration is not necessary during training.

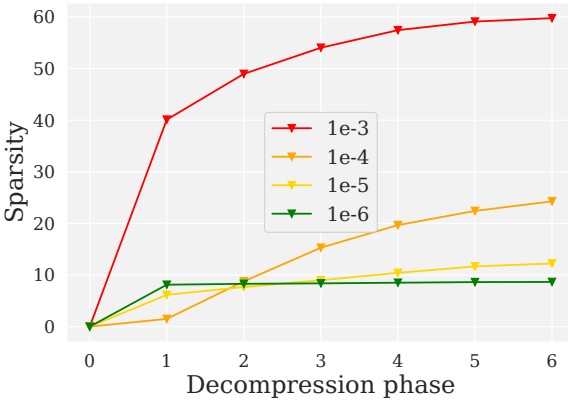

| Weight decay | Top-1 accuracy (%) |
|---|---|
| $10^{-6}$ | 70.52 |
| $10^{-5}$ | 73.54 |
| $10^{-4}$ | 74.90 |
| $10^{-3}$ | 68.57 |

Table L.9: Accuracy vs weight decay.

| Decompression sparsity | Top-1 accuracy (%) |
|---|---|
| 0 | 74.82 |
| 50 | 75.09 |
| 60 | 74.81 |
| 70 | 74.83 |
| 80 | 74.38 |

Figure L.6: Sparsity on decompression phases for 100-epoch AC/DC++ runs with varying values of weight decay. We point out that on decompression phases no sparsity is enforced.

Table L.10: Accuracy vs decompression sparsity.

## M   Loss landscape analysis

In order to get more insights into the reasons for the difficulty of optimizing sparse neural networks, we investigate two properties of the loss landscape. First, we measured *landscape sharpness at the end of the training*, defined as the maximal eigenvalue of the Hessian matrix, for all sparse training methods considered, various sparsities and number of training epochs and compared with the one of standard dense training. Second, we interpolated the training and validation loss between checkpoints obtained at intermediate steps throughout the 1000 epoch AC/DC run with 95% target sparsity.

The largest Hessian eigenvalue was estimated via the power iteration method based on Hessian-vector products using a customized version of the Eigenthings library (Golmant et al., 2018). To compute the largest Hessian eigenvalue we ran power iteration method (von Mises & Pollaczek-Geiringer, 1929) for 20 iterations using the modified version of the code from (Golmant et al., 2018) package. 20 iterations were usually enough for the iterations to converge. Hessian is computed only with respect to the non-zero weights. Technically, for a current estimate of a leading eigenvector $\mathbf{e}_i$ and sparsity mask $\mathbf{m}$ we compute $\nabla(\mathbf{m} \cdot \nabla(\mathbf{m} \cdot \mathbf{e}_i)) = \mathbf{H_m}$ which is equivalent to the hessian-vector product for a truly sparse model. In Figure **??**, we observe that, across all methods, sharpness increases with the length of the training run, indicating that sharper minima require extended training to be reached via SGD. Additionally, sharpness decreases with the increase of sparsity. All sparse training methods attain lower sharpness compared to the dense model. Models trained with AC/DC and RigL have slightly lower sharpness compared to models trained with GMP, presumably because the former two methods manage to reach flatter optima which are conjectured to have better generalization properties (Merity et al., 2017).

To examine mode connectivity behavior, in Figure M.7, we connected the checkpoint obtained on the $99^{th}, 199^{th}, \ldots 999^{th}$ epoch via piecewise-linear curves with 10 points between checkpoints. A notable observation is that all checkpoints are separated by a loss barrier, whose height is increasing with the number of epochs. Probably, this behavior is a manifestation of the progressive sharpening phenomenon (Cohen et al., 2022) where the model sharpness is increasing gradually with training until reaching the peak value and then

plateaus. Yet, for sparse models, the duration of the longest runs is not long enough to reach the sharpness plateau.

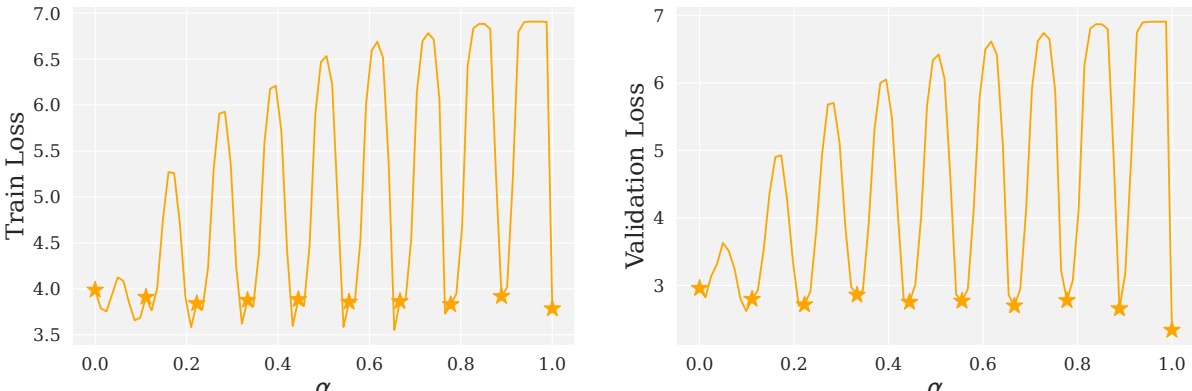

Figure M.7: Loss interpolation curves on the training (**left**) and validation (**right**) checkpoints of the 95% sparse 1000-epoch AC/DC model. $\alpha$ corresponds to the fraction of paths traversed from the first checkpoint to the last. Stars denote checkpoints between which loss is interpolated.

## N   Different sparsity patterns.

Since the ResNet models increase the number of channels with the decrease of feature map resolution, one would expect that bottom layers (those with more channels) should be pruned more aggressively compared to the one with less channels. Here our results are consistent with the known observations from literature that global sparsity achieves higher performance for the same sparsity. In addition, we have carried out experiments with the more hardware-friendly block-4 sparsity pattern.

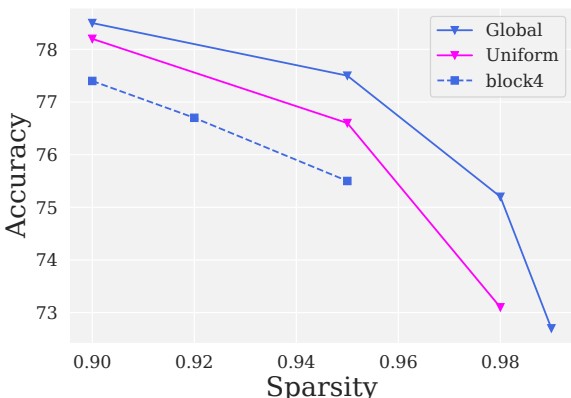

Figure N.8: Accuracy vs sparsity for different sparsity distributions. Block4 denotes global pruning with weights pruned in groups of 4.

## O   Transfer learning datasets for language models

In Table O.11 we provide a brief summary of datasets in the popular GLUE benchmark (Wang et al., 2018). Following previous work (Sanh et al., 2020; Zafir et al., 2021; Kurtic & Alistarh, 2022; Kurtic et al., 2022; Zhang et al., 2022; Huang et al., 2021), we exclude WNLI dataset from consideration in all experiments.

| Dataset | Train/Test Examples | Accuracy Metric |
|---|---|---|
| RTE (Wang et al., 2018) | 2.5k / 3k | Accuracy |
| QNLI (Wang et al., 2018) | 105k / 5.4k | Accuracy |
| MRPC (Dolan & Brockett, 2005) | 3.7k / 1.7k | F1 score and Accuracy |
| SST-2 (Socher et al., 2013) | 67k / 1.8k | Accuracy |
| CoLA (Warstadt et al., 2019) | 8.5k / 1k | Matthews correlation coefficient |
| STS-B (Cer et al., 2017) | 7k / 1.4k | Pearson and Spearman correlation |
| MNLI (Williams et al., 2017) | 393k / 20k | Matched (m) and mismatched (mm) accuracy |
| QQP (Wang et al., 2018) | 364k / 391k | Accuracy and F1 score |

Table O.11: Datasets used as downstream tasks for transfer learning with language models.

# P   Additional plots

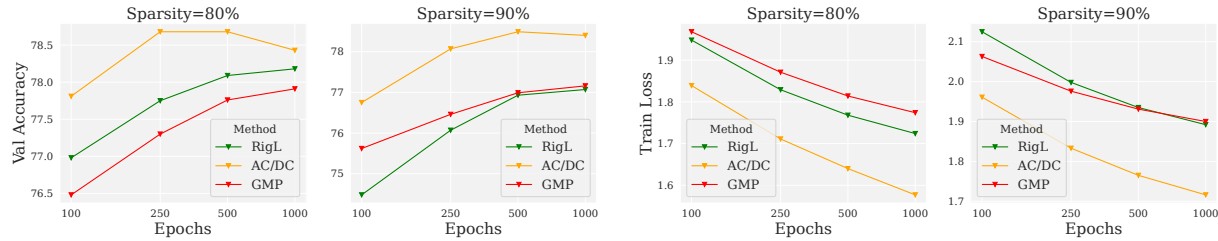

Figure P.9: (**left**) Validation accuracy on ImageNet-1k vs number of epochs for different sparse training methods. (**right**) Training loss on ImageNet-1k vs number of epochs for different sparse training methods.