# OpenReview forum: "Accurate Neural Network Pruning Requires Rethinking Sparse Optimization"
_TMLR — Accepted by TMLR_

### Review · Reviewer_nRmi · 2024-01-18

**Summary Of Contributions:**

This paper presents a comprehensive overview of the sparse training. In particular, the authors point out that current sparse training algorithms that adopt the hyper-parameters of dense training will generally lead to undertraining of models. To this end, the authors try to extend the training pipeline by 5x or 10x, and successfully achieve better performance on standard vision tasks. The authors further examine several properties related to the sparsity, such as the structured sparse masks, the effect of weight decays, and the loss landscapes. Finally, the authors have studied the behaviors of sparsity under a more typical transfer scenario, i.e. language modeling, and have successfully improved the performance of sparse transfer.

**Audience:**

Yes

**Broader Impact Concerns:**

-

**Claims And Evidence:**

Yes

**Requested Changes:**

- Probably better organize the paper, and provide better support for sections in the paper.

**Strengths And Weaknesses:**

Strengths:
- First of all, the topic is valuable as sparse models are getting more and more powerful (such as those SMoE models)
- Second, massive amount of experiments are provided in this paper to support the authors' claim. I feel this is a very strong empirical paper regarding the sparse training.
- A lot of design choices are covered.

Weakness:
- Essentially there is no additional technique proposed in this paper except for training for longer time. If the authors can provide some principled techniques to deal with the general under-training problem that would be better.
- While a lot of experiments are provided, I still think some parts are loosely connected, or at least need a better connection between paragraphs and paragraphs. For instance, the sudden present of BERT might need better explanation, and probably should not just mentioning "Motivated by our findings for computer-vision models in previous sections,".

---

> ### Author Response · Authors · 2024-03-08
> **Response**
>
> Thank you for your helpful comments and for recognizing the relevance and thoroughness of our work. Please see our responses to your weaknesses and suggestions below.
>
> **Weaknesses**
>
> * The aim of our work was to demonstrate that, with current pruning methods, **extended training** and **additional hyperparameter** tuning are required to produce highly sparse, accurate models. We also provide practical guidelines for doing so and use these to demonstrate state-of-the-art performance. Further, we believe that our emphasis on longer training is a valuable contribution in its own right, as, currently, newly proposed training methods still largely stay within the epoch budget of dense training. It’s also possible to interpret the reviewer’s feedback to mean that it would be valuable to instead focus on producing highly sparse accurate models within a more constrained training budget. Our work is** not** meant to demonstrate or suggest the *impossibility* of doing so; in fact, we hope that it will *encourage* work in that direction. We have extended the discussion/future work section to highlight this point.
>
> * We have re-written the paper to provide a smoother transition between the sections, and to better balance the content between the vision and language applications. Our goal in the paper is to investigate the changes to sparse training algorithms to enable much higher sparsity. As the language modeling setup is different (language models are more frequently finetuned from foundational models), the transition can seem a bit abrupt; we hope that it is less so now.
>
> **Suggestions**
>
> We agree with your suggestions and have reworked the paper to have better motivated transitions.

---

### Review · Reviewer_iVXw · 2024-01-19

**Summary Of Contributions:**

This paper mainly focuses on learning high-accurate and high-sparse models. It notices that few work has explore the interaction between sparsity and standard stochastic optimization techniques. Thus, in this work, it proposes new approaches for mitigating the issue. The proposed method achieves state-of-the-art performance.

**Audience:**

No

**Claims And Evidence:**

No

**Requested Changes:**

- The writting of the paper can be further improved;

- The definition of some concepts, such as _label-independent_ , are not well defined, which is confusing.

- More information can be added in figures.

- Too many content regarding experimental settings. It would be better to concisely mention these content and provide more intuition about the goal of the experiments conducted.

**Strengths And Weaknesses:**

__Pros:__
- The results are good.


__Cons:__
- Why the phenomenon that the accuracy and loss of a sparse model fails to saturate means undertraining? Do you consider this phenomenon from other perspectives? If not, from my perspective, I think it is normal since sparsity undermines the original structure of models and the connections among parameters. Thus, I do not think the motivation in this paper is strong.

- The two observations mentioned in page 6 seems trivial. What the role does observation 1 play in this paper? It seems a common sense.

- The curves in Fig 1 center and right seems with with the same shape.

- Section 3 is not compact. There are many trivial contents which are little bit confusing.

- This paper repeatedly mentions that sparse model requires more training episode in Section 3. From my perspective, I think it is a normal phenomenon and one experiment is enough. Besides, I am not sure about the conclusion in Section 3.3.1. Why the results 'reinforce the point that sparse training method saturate slower'?  Why is this important?

- I did not find the so called method mentioned in the abstract? Could you make it more clear?

---

> ### Author Response · Authors · 2024-03-08
> **The response**
>
> Thank you for your feedback! We have tightened the main body of the paper in a way that we hope focuses more on the important points rather than experimental details.
>
> **Weaknesses**
>
> * **Undertraining** means that the model is not trained long enough to achieve the best performance on held-out data, i.e the neural network still benefits from extended training. We have clarified this throughout the paper. Generally, the motivation for the paper is that, even though other works have already noticed that extended training improves the accuracy of sparse networks, this has not been systematically studied. As a consequence, today, most pruning-focused works still measure the method’s effectiveness in the standard 100-epoch training regime, which we show to be grossly suboptimal for sparse networks. In more detail, our work provides a systematic investigation of this phenomenon, and provides guidance on why it occurs, and when a sparse network is sufficiently trained (when the training loss or prediction entropy roughly matches that of the dense model). Additionally, we demonstrate new state-of-the-art results that we hope will encourage additional work on improving sparse training dynamics. It also demonstrates something that we believe is far from trivial - that while it is intuitive that sparse training is harder, we don’t yet understand the limits of what is possible. For instance, we are the first to train 98% sparse ImageNet ResNet50 models that  come within 1.5% of the dense model accuracy–previously, this threshold was around 90% sparsity, i.e. almost an order of magnitude lower.
> * **Label-independent**, as the name suggests, means that this quantity does not depend on the *target label* and therefore can be computed without any need for having access to labels. The **prediction entropy** involves only the prediction of the model and no label of the ground truth class. However, per this reviewer’s feedback, we have moved this plot to the appendix and now concentrate only on the training loss and validation accuracy in the body.
> * We broadly agree with this point, and have moved one of the plots to the appendix. At the same time, the curves have a similar shape, but represent different quantities. Training loss is a measure of convergence on the training data, and the validation cross-entropy is a measure of the model (epistemic) uncertainty. It is natural to expect that these show close behavior.
> * Thank you for pointing this out. We have reorganized this section and moved much of the detailed analysis to the appendix, which we believe makes the paper more readable.
> * We demonstrate that this phenomenon is consistent across different sparse training methods and sparsities. Moreover, with the increase of sparsity, undertraining becomes more pronounced.
> * We agree that we do not present a new method, but rather refine an existing one, AC/DC, thus showing that the accuracy/sparsity tradeoff can be pushed further via extended training. This is also reflected in the name of our refined method - **AC/DC++**. We amended the manuscript to clarify this point.

---

### Review · Reviewer_UEcX · 2024-02-24

**Summary Of Contributions:**

The paper studies the challenges in the optimisation of sparse deep neural networks, both during pre-training and transfer settings, analysing different existing algorithms (AC/DC, RigL, GMP, …) and proposing some modifications to improve the quality of the resulting sparse models at different sparsity levels. The work includes a wide range of experiments: Most of them were done on image recognition tasks during pre-training of the sparse models (section 3), but section 4 analyses sparse transfer learning for language models. The two main discoveries from the study are that (1) sparse models need longer pre-training/fine-tuning, (2) using dense first and last layers makes sparse pre-training in vision, (3) while transferring language models it’s better to sparsify the layers gradually, from top (closer to head) to bottom, while fine-tuning the model.

**Audience:**

Yes

**Broader Impact Concerns:**

No major concerns from my perspective.

**Claims And Evidence:**

Yes

**Requested Changes:**

- I did not quite understand how the experiments in section 3.7 are performed. Could you provide more details on how you compute the eigenvalues of the hessian matrix? The hessian matrix is not even well defined when one considers the mask, since it is a discrete (binary) variable, with no such thing as “gradients”. This is really important, since I could not really follow this section, which could greatly improve my final recommendation.
- Given my feedback in the weaknesses section above, I would suggest to trim down some experiments sections, or move them to the appendix. As I mentioned, some experiments yield quite unsurprising results. This is just a minor suggestion, but I think would make the reading shorter and the understanding of main conclusions simpler.

**Strengths And Weaknesses:**

**Strengths**
- The paper contains a detailed section dedicated to describe the state-of-the-art methods dealing with sparse deep neural networks. This is very important in all works, but it’s crucial for a study paper like this that compares many existing methods. I found this section very useful while reviewing the manuscript.
-  The paper does a thorough analysis of the pre-training of sparse deep neural networks. Section 3 contains a wide range of very relevant experiments (other maybe redundant, see weaknesses below).
- Table 1 contains a wide set of results comparing the proposed modifications to AC/DC with many previously published works. This is a fantastic and quite complete results table of recent and relevant works in sparse deep neural networks.
- I like very much the writing of the paper


**Weaknesses**
- Something that I’m missing in the related works section is a review of related works dealing with the _optimisation_ perspective of sparse models. In section 3.1, the paper mentions that the optimisation of sparse models is NP-Hard, even on simple regression models, but there is no mention to related works from the optimisation community. For instance, some works that deal with the sparse optimisation of a more simple problem (optimal transport, but with applications to sparse neural networks): Smooth and Sparse Optimal Transport (M. Blondel et al., 2017), Sparsity-Constrained Optimal Transport (T. Liu et al., 2022).
- Given that the optimisation of sparse models is NP-Hard, the observation that longer training benefits sparse models (assuming no data size constraints) is not surprising at all. This reduces a lot of the amount of information that lots of experiments contribute (it also diminishes the value of the main contribution), in my opinion.
- Section 3.6 also devotes a lot of text and experiments to study a well-known fact: “once a channel is completely zeroed out, it will continue to receive zero gradients”. This is a simple result of computing the gradients of a linear layer w.r.t params and inputs, and it’s (one of) the reason(s) why neural networks of more than 1 layer are initialised with small weights close to zero _but not exactly zero_. So, it’s not surprising at all that this happens more frequently when weight decay is applied more heavily.
- The conclusion of the sparse decompression subsection is a bit misleading. The paper claims: “We observe that the performance is almost una ffected up to 80% decompression sparsity, showing that full mask exploration is not necessary during training”. The observation does not strictly support that conclusion: the fact that the performance doesn’t improve does not directly imply that full mask exploration is unnecessary, it could just show that the decompression approach is suboptimal. In fact, given the fundamental NP-Hardness of the problem, one could argue that mask exploration is indeed _very_ important.

---

> ### Author Response · Authors · 2024-03-08
> **The response**
>
> Thank you for your feedback and comments! Please see our responses to your weaknesses and suggestions below.
>
> **Weaknesses**
>
> * Thanks for pointing out related work from the optimization literature. In the related work section, we covered primarily the literature on sparse training of neural networks, which is the focus of our work. However, we agree that the literature dealing with sparse optimal transport is worth being highlighted, and we will cover these works in the revised version of our work.
> * The observation that longer training leads to significantly improved accuracy may not appear surprising, given NP-Hardness. However, we believe that it is valuable in context since: 1) this observation has been largely **overlooked** in the literature–the vast majority of existing work uses **standard recipes**; and 2) the degree to which longer training helps–e.g. allowing us to reach extremely high sparsities in the 98% range with high accuracy–may be seen as surprising. As such, our message is that the sparse models obtained via sparse-training / gradual pruning are often **not trained long enough**, and the performance can be pushed further.
> * The observation that a zero channel will stay at zero is never claimed to be novel–it is, as you state, a basic fact. However, we do highlight its non-trivial interactions with two other components: weight decay (which brings weights consistently towards zero) and the periodic pruning mechanism (which “catches” channels which have not received enough gradients). This interaction is not straightforward: for instance, higher weight decay is indeed likely to increase the probability of a channel to be eliminated, but it doesn’t follow immediately from its definition, as it penalizes the norm of the weights rather than induces sparsity.
> * We believe we are in agreement on this point, and we will rephrase for clarity.
> Specifically, the outcome of the sparse decompression experiment shows that there is no difference in performance between the AC/DC that is allowed to reintroduce any weight and AC/DC that can reintroduce only 20% of weights, i.e. significantly more constrained in mask exploration. The observation is that even being allowed to regrow any weight, the AC/DC technique doesn’t find better candidates for regrowth during decompression.
> We will clarify this point to state that we mean this about our method, not any dynamic sparse training method in general.
>
> **Requested changes**
>
> * Eigenvalues are computed via the **power iteration** method, which requires only the computation of hessian-vector products. We do not compute the elements of a Hessian matrix corresponding to the masked elements, and treat them if they were absent from the model. In practice, we achieve this as follows: we compute $\nabla (\nabla (w * m) * m)$ treating the network of interest as a sparse one. For a sparse network, Hessian is defined in the *same* way as for a dense one.
> * We agree with your suggestions, and we have shortened the aforementioned experiments section to help readability.

---

### Author Response · Authors · 2024-03-08
**General response and summary of changes**

We thank the reviewers for their thoughtful feedback. We appreciate that the reviewers noted the importance of the field of sparse training, as well as the paper’s strong empirical results and thorough survey of existing pruning methods.

At the same time, reviewers pointed out that the detailed experiments presented in `Section 3` felt too extensive, perhaps obscuring the main point of the paper. Based on this feedback, we reorganized that entire section, moving many of the supporting experimental results to the appendix and concentrating the main body on our main goal, which is to thoroughly investigate the proper conditions for sparse training and transfer learning in image and text models.

We hope that the revision better highlights our contribution, which is, in a nutshell the first thorough overview of existing sparsification techniques for both vision and language from the perspective of maximizing the accuracy of highly-sparse models, showing that state-of-the-art results can be obtained by carefully balancing parameters such as number of epochs and weight decay, and highlighting the need for better optimization mechanisms focused on sparse training (as suggested by the title).

Please see below for responses to the reviewers’ individual feedback. We have uploaded an updated draft of the paper that reflects the feedback and changes.

---

### Decision · Action_Editor_G1Bu · 2024-05-02

**Recommendation:** Accept as is

**Comment:**

While reviewers are unconvinced about the novelty and significance of the work, this is a good example where TMLR's primary criteria of evidence matching claims comes in. Based on this criteria and the unanimous agreement by reviewers that the claims match the evidence, I see no reason not to accept the paper.

**Audience:**

Yes

**Claims And Evidence:**

Yes, while reviewers are generally negative on the novelty and size of the contributions, they are unanimous that the claims match the evidence.